 SciPost Phys. Lect.Notes 28 (2021)

# Probabilistic theories and reconstructions of quantum theory

**Markus P. Müller**[1,2*]

**1** Institute for Quantum Optics and Quantum Information,
Austrian Academy of Sciences, Boltzmanngasse 3, A-1090 Vienna, Austria
**2** Perimeter Institute for Theoretical Physics,
31 Caroline Street North, Waterloo, ON N2L 2Y5, Canada

* markus.mueller@oeaw.ac.at

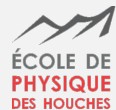

*Part of the Quantum Information Machines*
*Session 113 of the Les Houches School, August 2019*
*published in the Les Houches Lecture Notes Series*

## Abstract

These lecture notes provide a basic introduction to the framework of generalized probabilistic theories (GPTs) and a sketch of a reconstruction of quantum theory (QT) from simple operational principles. To build some intuition for how physics could be even more general than quantum, I present two conceivable phenomena beyond QT: super-strong nonlocality and higher-order interference. Then I introduce the framework of GPTs, generalizing both quantum and classical probability theory. Finally, I summarize a reconstruction of QT from the principles of Tomographic Locality, Continuous Reversibility, and the Subspace Axiom. In particular, I show why a quantum bit is described by a Bloch ball, why it is three-dimensional, and how one obtains the complex numbers and operators of the usual representation of QT.

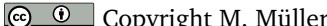

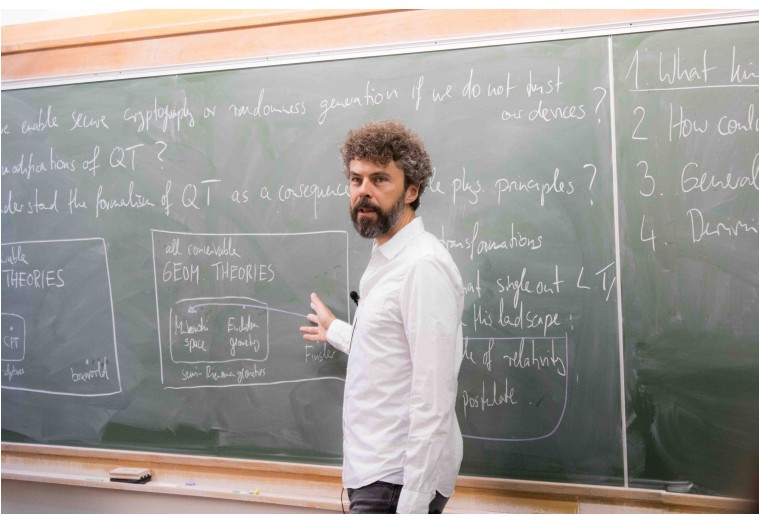

## 1   What kind of "quantum foundations"?

These lectures will focus on some topics in the foundations of quantum mechanics. When physicists hear the words "Quantum Foundations", they typically think of research that is concerned with the problem of *interpretation*: how can we make sense of the counterintuitive Hilbert space formalism of quantum theory? What do those state vectors and operators really tell us about the world? Could the seemingly random detector clicks in fact be the result of deterministic but unobserved "hidden variables"? Some of this research is sometimes regarded with suspicion: aren't those Quantum Foundationists asking questions that are ultimately irrelevant for our understanding and application of quantum physics? Should we *really* care whether unobservable hidden variables, parallel universes or hypothetical pilot waves are the mechanistic causes of quantum probabilities? Isn't the effort to answer such questions simply an expression of a futile desire to return to a "classical worldview"?

For researchers not familiar with this field, it may thus come as a surprise to see that large parts of Quantum Foundations research today are *not* primarily concerned with the interpretation of quantum theory (QT) — at least not directly. Much research effort is invested in proving rigorous mathematical results that shed light on QT in a different, more "operational" manner, which is motivated by quantum information theory. This includes research questions like the following:

(i) Is it possible to generate secure cryptographic keys or certified random bits even if we do not trust our devices?

(ii) Which consistent modifications of QT are in principle possible? Could some of these modifications exist in nature?

(iii) Can we understand the formalism of QT as a consequence of simple physical or information-theoretic principles? If so, could this tell us something interesting about other open

problems of physics, like e.g. the problem of quantum gravity?

Question (i) shows by example that some of Quantum Foundations research is driven by ideas for technological applications. This is in some sense the accidental result of a fascinating development: it turned out that such "technological" questions are surprisingly closely related to foundational, conceptual ("philosophical") questions about QT. To illustrate this surprising relation, consider the following foundational question:

(iv) Could there exist some hidden variable (shared randomness) $\lambda$ that explains the observed correlations on entangled quantum states?

Question (iv) is closely related to question (i). To see this, consider a typical scenario in which two parties (Alice and Bob) act with the goal to generate a secure cryptographic key. Suppose that Alice and Bob hold entangled quantum states and perform local measurements, yielding correlated outcomes which they can subsequently use to encrypt their messages. Could there be an eavesdropper (say, Eve) somewhere else in the world who can spy on their key? Intuitively, if so, then we could consider the key bits that Eve learns as a hidden variable $\lambda$: a piece of data ("element of reality", see Ekert, 1991 [32]) that sits somewhere else in the world and, while being statistically distributed, can be regarded as determining Alice's and Bob's outcomes. But Bell's Theorem (Bell, 1964 [13]) tells us that the statistics of some measurements on some entangled states are *inconsistent* with such a (suitably formalized) notion of hidden variables, unless those variables are allowed to exert nonlocal influence. This guarantees that Alice's and Bob's key is secure in such cases, as long as there is no superluminal signalling between their devices and Eve. The conclusion holds even if Alice and Bob have no idea about the inner workings of their devices — or, in the worst possible case, have bought these devices from Eve. This intuition can indeed be made mathematically rigorous, and has led to the fascinating field of *device-independent cryptography* (Barrett, Hardy, Kent (2005) [10]) and *randomness expansion* (Colbeck (2006) [25], Colbeck and Kent (2011) [26], Pironio et al. (2010) [69]).

The preconception that Quantum Foundations research is somehow motivated by the desire to return to a classical worldview is also sometimes arising in the context of question (ii) above. It is true that the perhaps better known instance of this question asks whether QT would somehow break down and become classical in the macroscopic regime: for example, spontaneous collapse models (Ghirardi, Rimini, and Weber (1986) [36], Bassi *et al.* (2013) [12]) try to account for the emergence of a classical world from quantum mechanics via dynamical modifications of the Schödinger equation. However, a fascinating complementary development in Quantum Foundations research — the one that these lectures will be focusing on — is to explore the exact opposite: *could nature be even "more crazy" than quantum?* Could physics allow for even stronger-than-quantum-correlations, produce more involved interference patterns than allowed by QT, or enable even more magic technology than what we currently consider possible? If classical physics is an approximation of quantum physics, could quantum physics be an approximation of something even more general?

As we will see in the course of these lectures, the answer to these questions is "yes": nature *could* in principle be "more crazy". The main insight will be that QT is just one instance of a large class of *probabilistic theories*: theories that allow us to describe probabilities of measurement outcomes and their correlations over time and space. Another example is "classical probability theory" (CPT) as defined below, but there are many other ones that are equally consistent.

As we will see, not only is there a simple and beautiful mathematical formalism that allows us to describe all such theories, but the new approach to QT "from the outside" provides a very illuminating perspective on QT itself: it allows us to understand which features are uniquely quantum and which others are just general properties of probabilistic theories. Moreover, it

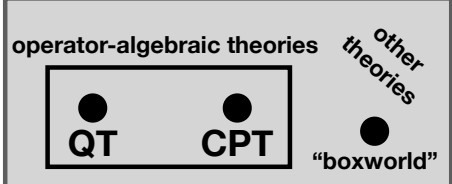
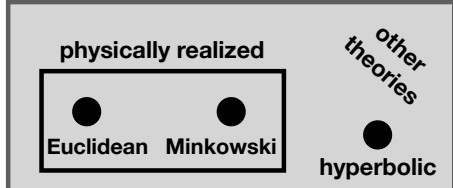

Figure 1: **Left:** the "landscape" of probabilistic theories. QT is for quantum theory and CPT for classical probability theory (as defined later). **Right:** as a suggestive analogy (see main text), the "landscape of theories of (spacetime) geometry".

gives us the right mathematical tools to describe physics in the, broadly construed, "device-independent" regime where all we want to assume is just a set of basic physical principles. Making sparse assumptions is arguably desirable when approaching unknown physical terrain, which is why some researchers consider some of these tools (and generalizations thereof) as potentially useful in the context of quantum gravity (Oreshkov *et al.* (2012) [64]), and, as we will see, for fundamental experimental tests of QT.

Even more than that: as we will see in these lectures, it is possible to write down a small set of physical or information-theoretic postulates that singles out QT uniquely within the landscape of probabilistic theories. We will be able to reconstruct the full Hilbert space formalism from simple principles, starting in purely operational terms *without* assuming that operators, state vectors, or complex numbers play any role in it. Not only does this shed light on the seemingly ad-hoc mathematical structure of QT, but can also indirectly give us some hints on how we might want to interpret QT.

There is a historical analogy to this strategy that has been described by Clifton, Bub, and Halvorson (2003) [18] (see, however, Brown and Timpson (2006) [17] for skeptic remarks on this perspective). Namely, the development of Einstein's theory of special relativity can be understood along similar lines: there is a landscape of "theories of (spacetime) geometry", characterized by an overarching, operationally motivated mathematical framework (perhaps that of semi-Riemannian geometry). This landscape contains, for example, Euclidean geometry (a very intuitive notion of geometry, comparable to CPT in the probabilistic landscape) and Minkowski geometry (less intuitive but physically more accurate, comparable to QT in the probabilistic landscape). Minkowski spacetime is characterized by the Lorentz transformations, which have been historically discovered in a rather ad hoc manner — simply postulating these transformations should invite everybody to ask "why?" and "could nature have been different"? But Einstein has shown us that two simple physical principles single out Minkowski spacetime, and thus the Lorentz transformations, uniquely from the landscape: the relativity principle and the light postulate. This discovery is without doubt illuminating by explaining "why" the Lorentz transformations have their specific form, and it has played an important role in the subsequent development of General Relativity.

In these lectures, we will see how a somewhat comparable result can be obtained for QT, and we will discuss how and why this can be useful. But before going there, we need to understand how a "generalized probabilistic theory" can be formalized. And even before doing so, we need to get rid of the widespread intuition that all conceivable physics must either be classical or quantum, and build some intuition on *how physics could be more general than quantum*.

## 2  How could physics be more general than quantum?

Everybody can take an existing theory and modify it arbitrarily; but the art is to find a modification that is self-consistent, physically meaningful, and consistent with other things we know about the world.

That these desiderata are not so easy to satisfy is illustrated by Weinberg's (1989) [85] attempt to introduce nonlinear corrections to quantum mechanics. QT predicts that physical quantities are described by Hermitian operators ("observables") $A$, and their expectation values are essentially bilinear in the state vector, i.e. $\langle A \rangle = \langle \psi | A | \psi \rangle$. (This property is closely related to the linearity of the Schrödinger equation.) Weinberg decided to relax the condition of bilinearity in favor of a weaker, but arguably also natural condition of homogeneity of degree one, and explored the experimental predictions of the resulting modification of quantum mechanics.

However, shortly after Weinberg's paper had appeared, Gisin (1990) [37] pointed out that this modification of QT has a severe problem: it allows for faster-than-light communication. Gisin showed how local measurements of spacelike separated parties on a singlet state allows them to construct a "Bell telephone" with instantaneous information transfer within Weinberg's theory. Standard QT forbids such information transfer, because bilinearity of expectation values implies (in some sense — we will discuss more details of this "no-signalling" property later) that different mixtures with the same local reduced states cannot be distinguished. Superluminal information transfer is in direct conflict with Special Relativity, showing that QT is in some sense a very "rigid" theory that cannot be so easily modified (see also Simon *et al.*, 2001 [77]).

This suggests to search for modifications of QT not on a formal, but on an *operational* level: perhaps a more fruitful way forward is to abandon the strategy of direct modification of any of QT's *equations*, and instead to reconsider the basic *framework* which we use to describe simple laboratory situations. GPTs constitute a framework of exactly that kind. They generalize QT in a consistent way, and do so without introducing pathologies like superluminal signalling.

To get an intuition for the basic assumptions of the GPT framework, let us first discuss two examples of potential phenomena that would transcend classical *and* quantum physics: superstrong nonlocality and higher-order interference.

### 2.1  Nonlocality beyond quantum mechanics

Consider the situation in Figure 2. In such a "Bell scenario", we have two agents (usually called Alice and Bob) who each independently perform some local actions. Namely, Alice holds some box to which she can input a freely chosen variable $x$ and obtain some outcome $a$. Similarly, Bob holds a box to which he can input some freely chosen variable $y$ and obtain some outcome $b$. Alice's and Bob's boxes may both have interacted in the past, so that they may have become statistically correlated or (in quantum physics) entangled. This will in general lead to correlations between Alice's and Bob's outcomes.

While more general scenarios can be studied, let us for simplicity focus on the case that there are two agents (Alice and Bob) who can choose between two possible inputs $x, y \in \{0, 1\}$ and obtain one of two possible outcomes $a, b \in \{-1, +1\}$. In quantum information jargon, we are on our way to introduce the $(2, 2, 2)$-Bell correlations, where $(m, n, k)$ denotes a scenario with $m$ agents who each have $n$ possible inputs and $k$ possible outcomes. The resulting statistics is thus described by a probability table (often called "behavior")

$$P(a, b | x, y),$$

i.e. the conditional probability of Alice's and Bob's outcomes, given their choices of inputs. It is clear that these probabilities must be non-negative and $\sum_{a,b} P(a, b | x, y) = 1$ for all $x, y$ (we

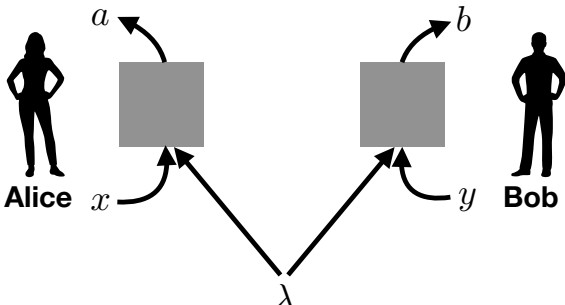

Figure 2: Schematic figure of a Bell scenario. Within any physical theory (classical physics, quantum physics, or other), we can imagine laboratory situations in which the causal structure is specifically as depicted (in particular, Alice and Bob cannot communicate). Regardless of the interpretation of "probability", we can talk about situations in which there is data to be chosen independently ($x$ and $y$) and recorded ($a$ and $b$) such that it is meaningful to talk about the probabilities of the recordings, given the choices. Data from the past that may influence the devices will be labelled $\lambda$. Physical theories differ in the set of probability tables ("correlations") that they allow in principle in this scenario.

will assume this in all of the following), but further constraints arise from additional physical assumptions.

Let us first assume that the scenario is described by **classical probability theory** — perhaps because we are in the regime of classical physics or every-day life. Then we can summarize the causal past of the experiment — everything that has happened earlier and that may have had some influence on the experiment, directly or indirectly — into some variable $\lambda$. Shortly before Alice and Bob input their choices into the boxes, the variables $x, y, \lambda$ are in some (unknown) configuration, distributed according to some probability distribution $P$. Furthermore, the outcomes $a$ and $b$ are random variables; in the formalism of probability theory, they must hence be functions of $x, y$ and $\lambda$, i.e. $a = f_A(x, y, \lambda)$, $b = f_B(x, y, \lambda)$. Recall that in probability theory, random variables are functions on the sample space; and the sample space, describing the configuration of the world, consists of only $x, y$ and $\lambda$. We have simply made $\lambda$ big enough to contain everything in the world that is potentially relevant for the experiment.

But what if the boxes introduce some additional randomness, perhaps tossing coins to produce the outcome? In this case, the coin toss can be regarded as deterministic if only all the factors that influence the coin toss (properties of the coin, the surrounding air molecules etc.) are by definition contained in $\lambda$. (Or, alternatively, we simply regard the unknown state of the coin as a part of $\lambda$.) The "hidden variable" $\lambda$ may thus be a quite massive variable, and learning its value may be practically impossible. In other words, all randomness can, at least formally, be considered to result from the experiment's past (in physics jargon, the fluctuations of its initial conditions).

So far, our description is completely general and does not yet take into account the assumed *causal structure* of the experiment: assuming that $x$ and $y$ can be chosen freely amounts to demanding that their values are statistically uncorrelated with everything that has happened in the past, i.e. with $\lambda$. Furthermore, locality implies that $a$ cannot depend on $y$ and $b$ cannot depend on $x$. This means that the scenario must satisfy

$$P(x, y, \lambda) = P_X(x) \cdot P_Y(y) \cdot P_\Lambda(\lambda), \qquad a = f_A(x, \lambda), \ \ b = f_B(y, \lambda).$$

For a more detailed explanation of how and why the causal structure of the setup implies these assumptions, see e.g. the book by Pearl (2009) [66], or Wood and Spekkens (2015) [88]. These

assumptions are typically subsumed under the notion of "local realism", and readers who want to learn more about this are invited to consult more specialized references. A great starting point are the Quantum Foundations classes given by Rob Spekkens at Perimeter Institute; these can be watched for free on *http://pirsa.org*.

Note that $P(a, b|x, y, \lambda) = \delta_{a, f_A(x, \lambda)} \delta_{b, f_B(y, \lambda)} = P_A(a|x, \lambda) P_B(b|y, \lambda)$ (with $\delta$ the Kronecker delta). Hence, by the chain rule of conditional probability,

$$P(a, b|x, y) = \sum_{\lambda \in \Lambda} P_A(a|x, \lambda) P_B(b|y, \lambda) P_\Lambda(\lambda). \tag{1}$$

What we have thus shown is that any probability table in classical physics that is realizable within the causal structure as depicted in Figure 2 must be *classical* according to the following definition:

**Definition 1.** *A probability table $P(a, b|x, y)$ is* classical *if there exists a probability space $(P, \Omega, \Sigma)$ with $P = P_X \cdot P_Y \cdot P_\Lambda$ some product distribution, $\Omega = X \times Y \times \Lambda$, where $X = Y = \{0, 1\}$ and $\Lambda$ arbitrary, such that Eq. (1) holds. If this is the case, then we call $(P, \Omega, \Sigma)$ a* hidden-variable model *for the probability table.*

*Denote by $\mathcal{C}_{2,2,2}$ the set of all classical probability tables.*

Instead of assuming that $\Lambda$ is a finite discrete set, we could also have allowed a more general measurable space like $\mathbb{R}^n$, but here this would not change the picture because the sets of inputs and outcomes are discrete and finite (in other words, considering only finite discrete $\Lambda$ is no loss of generality here).

In the derivation above, we have obtained a model for which $P_A(a|x, \lambda)$ and $P_B(b|y, \lambda)$ are deterministic, i.e. take only the values zero and one. But even *without* this assumption, probability tables that are of the form (1) can be realized within the prescribed causal structure according to classical probability theory: one simply has to add local randomness that makes the response functions $P_A$ and $P_B$ act nondeterministically to their inputs. Thus, we do not need to postulate in Definition 1 that $P_A$ and $P_B$ must be deterministic.

It is self-evident that the classical probability tables satisfy the **no-signalling conditions** (Barrett 2007 [11]): that is, $P_A(a|x, y) := \sum_b P(a, b|x, y)$ is independent of $y$, and $P_B(b|x, y) := \sum_a P(a, b|x, y)$ is independent of $x$. This means that Alice "sees" the local marginal distribution $P_A(a|x, y) = P_A(a|x) = \sum_{\lambda \in \Lambda} P_A(a|x, \lambda) P_\Lambda(\lambda)$ if she does not know what happens in Bob's laboratory, *regardless* of Bob's choice of input $y$ (and similarly with the roles of Alice and Bob exchanged). If this was *not* true, then Bob could signal to Alice simply by choosing the local input to his box. The causal structure that we have assumed from the start precludes such magic behavior.

We can reformulate what we have found above in a slightly more abstract way that will become useful later. Note that we have found that the classical behaviors are exactly those that can be expressed in the form (1) with $P_A$ and $P_B$ deterministic (if we want). Thus, we have shown that

**Lemma 2.** *A probability table is* classical *if and only if it is a convex combination of* deterministic non-signalling *probability tables.*

Here and in the following, we use some basic notions from *convex geometry* (see e.g. the textbook by Webster 1994 [86]). If we have a finite number of elements $x_1, \ldots, x_n$ of some vector space (for example probability distributions, or vectors in $\mathbb{R}^m$), then another element $x$ is a *convex combination* of these if and only if there exist $p_1, \ldots, p_n \geq 0$ with $\sum_{i=1}^n p_i = 1$ and $\sum_{i=1}^n p_i x_i = x$. Intuitively, we can think of $x$ as a "probabilistic mixture" of the $x_i$, with weights $p_i$. Indeed, the right-hand side of (1) defines a convex combination of the

$P^\lambda(a, b|x, y) := P_A(a|x, \lambda)P_B(b|y, \lambda)$. These are probability tables that are deterministic (take only values zero and one) and non-signalling (in fact, uncorrelated).

This reformulation has intuitive appeal: classically, all probabilities can consistently be interpreted as arising from *lack of knowledge*. Namely, we can put everything that we do not know into some random variable $\lambda$. If we knew $\lambda$, we could predict the values of all other variables with certainty.

Quantum theory, however, allows for a different set of probability tables in the scenario of Figure 2: instead of a joint probability distribution, we can think of a (possibly entangled) quantum state that has been distributed to Alice and Bob. The inputs to Alice's box can be interpreted as measurement choices (e.g. the choice of angle for a polarization measurement), and the outcomes can correspond to the actual measurement outcomes. This leads to the following definition:

**Definition 3.** *A probability table $P(a, b|x, y)$ is* quantum *if it can be written in the form*

$$P(a, b|x, y) = \text{tr}\left[\rho_{AB}(E_x^a \otimes F_y^b)\right],$$

*with $\rho_{AB}$ some density operator on the product of two Hilbert spaces $\mathcal{H}_A \otimes \mathcal{H}_B$, measurement operators $E_x^a, F_y^b \geq 0$ (i.e. operators that are positive semidefinite) and $E_x^{-1} + E_x^{+1} = \mathbf{1}_A$ as well as $F_y^{-1} + F_y^{+1} = \mathbf{1}_B$ for all $x$ and all $y$.*
*Denote by $\mathcal{Q}_{2,2,2}$ the set of all quantum probability tables.*

For our purpose, we will ignore some subtleties of this definition. For example, the state $\rho_{AB}$ can, without loss of generality, always be chosen pure, $\rho_{AB} = |\psi\rangle\langle\psi|_{AB}$, and the measurement operators can be chosen as projectors (see e.g. Navascues *et al.*, 2015 [61]). We will restrict our considerations to finite-dimensional Hilbert spaces; for the subtleties of the infinite-dimensional case, see e.g. Scholz and Werner, 2008 [74], and Ji et al., 2020 [49].

**Lemma 4.** *Here are a few properties of the classical and quantum probability tables:*

(i) *Both $\mathcal{C}_{2,2,2}$ and $\mathcal{Q}_{2,2,2}$ are* convex *sets, i.e. convex combinations of classical (quantum) probability tables are classical (quantum).*

(ii) $\mathcal{C}_{2,2,2} \subset \mathcal{Q}_{2,2,2}$.

(iii) *Every $P \in \mathcal{Q}_{2,2,2}$ is non-signalling.*

(iv) $\mathcal{C}_{2,2,2}$ *is a* polytope, *i.e. the convex hull of a* finite number *of probability tables. However, $\mathcal{Q}_{2,2,2}$ is not.*

Let us not prove all of these statements here, but simply explain some key ideas. Property (i) can easily be proved directly. For (ii), note that classical probability theory can be embedded in a commuting subalgebra of the algebra of quantum states and observables. Property (iii) is also easy to prove directly, and shows that measurements on entangled quantum states cannot lead to superluminal information transfer. For (iv), the *convex hull* of some points in a vector space is defined as the set of all vectors that can be obtained as convex combinations of those points. But there is only a finite number of deterministic non-signalling probability tables, and thus we obtain the statement for $\mathcal{C}_{2,2,2}$ by using Lemma 2.

The fact that $\mathcal{C}_{2,2,2}$ is a *strict subset* of $\mathcal{Q}_{2,2,2}$ is a consequence of Bell's (1964) [13] theorem, and it can be demonstrated e.g. via the CHSH inequality (Clauser *et al.*, 1969 [24]): if $P$ is any probability table, denote by $\mathbf{E}_{x,y}(P)$ the expectation value of the product of outcomes $a \cdot b$ on choice of inputs $x, y$, i.e.

$$\mathbf{E}_{x,y}(P) := P(+1, +1|x, y) + P(-1, -1|x, y) - P(+1, -1|x, y) - P(-1, +1|x, y),$$

and consider the specific linear combination

$$\mathbf{E}(P) := \mathbf{E}_{0,0}(P) + \mathbf{E}_{0,1}(P) + \mathbf{E}_{1,0}(P) - \mathbf{E}_{1,1}(P).$$

Then the CHSH Bell inequality (exercise!) states that

$$-2 \le \mathbf{E}(P) \le 2 \qquad \text{for all } P \in \mathcal{C}_{2,2,2}.$$

However, there are quantum probability tables that violate this inequality. These can be obtained, for example, via projective measurements on singlet states (see e.g. Peres 2002 [67]). The largest possible violation is known as the *Tsirelson bound* (Tsirelson 1980 [81])

$$\max_{P \in \mathcal{Q}_{2,2,2}} \mathbf{E}(P) = 2\sqrt{2}.$$

In particular, we find that $\mathcal{C}_{2,2,2} \subsetneq \mathcal{Q}_{2,2,2}$: nature admits "stronger correlations" than predicted by classical probability theory — but still not "strong enough" for superluminal information transfer.

This simple insight has motivated Popescu and Rohrlich (1994) [71] to ask: *are the quantum probability tables the most general ones that are consistent with relativity*? In other words, if we interpret the no-signalling conditions as the minimal prerequisites for probability tables to comply with the causal structure of Figure 2 within relativistic spacetime, then is QT perhaps the most general theory possible under these constraints?

The (perhaps surprising) answer is *no*: there are probability tables that are not allowed by QT, but that are nonetheless non-signalling. An example is given by the "PR box"

$$P^{\mathrm{PR}}(a, b | x, y) := \begin{cases} \frac{1}{2} & \text{if } a \cdot b = (-1)^{xy} \\ 0 & \text{otherwise.} \end{cases}$$

It is easy to check that this defines a valid probability table which satisfies the no-signalling conditions. If the two inputs are $(x, y) = (1, 1)$ then the outcomes are perfectly anticorrelated; in all other cases, they are perfectly correlated. Thus

$$\mathbf{E}(P^{\mathrm{PR}}) = 4,$$

which is larger than the maximal quantum value of $2\sqrt{2}$ (the Tsirelson bound). Therefore $P^{\mathrm{PR}} \notin \mathcal{Q}_{2,2,2}$. But if we denote the set of all non-signalling probability tables by $\mathcal{NS}_{2,2,2}$, then $P^{\mathrm{PR}} \in \mathcal{NS}_{2,2,2}$. Thus, we have the set inclusions

$$\mathcal{C}_{2,2,2} \subsetneq \mathcal{Q}_{2,2,2} \subsetneq \mathcal{NS}_{2,2,2}.$$

Like the set of classical probability tables, $\mathcal{NS}_{2,2,2}$ turns out to be a polytope, with corners (extremal points) given by the deterministic non-signalling tables as well as eight "PR boxes", i.e. versions of $P^{\mathrm{PR}}$ where inputs or outputs have been relabelled. This leads to the picture in Figure 3.

Thus, we have found one possible way in which nature could be more general than quantum: *it could admit "stronger-than-quantum" Bell correlations*. Clearly, simply writing down the probability table $P^{\mathrm{PR}}$ does not tell us anything about a possible place in the world where these correlations would potentially fit: we do not have a *theory* that would predict these correlations to appear in specific experimental scenarios. However, the same can be said about bare abstract quantum states: simply writing down a singlet state, for example, does not directly tell us what this state is supposed to represent. We need to impose additional assumptions (e.g. that the abstract quantum bit corresponds to a polarization degree of freedom that reacts

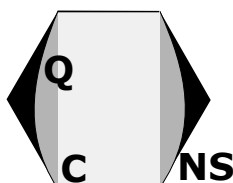

Figure 3: Schematic figure of the probability tables ("behaviors") that can be realized within classical probability theory ($\mathcal{C}$), within quantum physics ($\mathcal{Q}$), or within any non-signalling probabilistic theory ($\mathcal{NS}$). In the case of $(m, n, k) = (2, 2, 2)$, these convex sets are eight-dimensional; the 16 parameters in $P(a, b|x, y)$ are reduced by the normalization and no-signalling equalities to 8 free parameters. Both $\mathcal{C}$ and $\mathcal{NS}$ are polytopes, and the extremal points of $\mathcal{C}$ (which are also extremal points of $\mathcal{Q}$ and $\mathcal{NS}$) are the deterministic non-signalling probability tables. In contrast to $\mathcal{C}$ and $\mathcal{NS}$, the quantum set $\mathcal{Q}$ is not a polytope.

to spatial rotations in certain ways) in order to extract concrete predictions for specific experiments.

**Further reading**. The insight above has sparked a whole new field of research, asking "why" nature does not allow for stronger-than-quantum non-signalling correlations. One short answer is of course this: because physics is quantum. But simply pointing to the fact that the theories of physics that we have today are all formulated in terms of operator algebras and Hilbert spaces does not seem like a particularly insightful answer. Instead, the hope was to find *simple physical principles* that would explain, without direct reference to the quantum formalism, why nonlocality ends at the quantum boundary. An excellent overview on this research is given in (Popescu, 2014 [70]). Several physical principles have been discovered over the years which imply *part* of the quantum boundary, some of them including the Tsirelson bound: for example, some stronger-than-quantum correlations would trivialize communication complexity (van Dam 2013 [84] — published 8 years after the preprint on arXiv.org), violate information causality (Pawlowski *et al.*, 2009 [65]), or have an ill-defined classical limit (Navascués and Wunderlich, 2009 [62]). However, the discovery of *almost quantum correlations* (Navascués *et al.*, 2015 [61]), a natural set of correlations slightly larger than $\mathcal{Q}$ that satisfies all known reasonable principles, has severely challenged this particular research direction. An *exact* characterization of the quantum set, however, is achieved by the complementary program of reconstructing QT (not just its probability tables, but the full sets of states, transformations and measurements) within the framework of generalized probabilistic theories. This will be the topic of the last part of these lectures. In the case of $(m, n, k) = (2, 2, 2)$, the set $\mathcal{Q}$ can also be exactly characterized in terms of the detectors' local responses to spatial symmetries (Garner, Krumm, Müller, 2020 [34]).

## 2.2 Higher-order interference

Another way in which nature could be more general than quantum is in the properties of interference patterns that are generated by specific experimental arrangements. In 1994, Sorkin [79] proposed a notion of "order-$n$ interference" which contains "no interference at all" as its $n = 1$ case (as in classical physics), and quantum interference as the $n = 2$ case. In principle, however, nature could admit interference of order 3 or higher, and these potential beyond-quantum phenomena can be tested experimentally (up to a small caveat to be discussed below).

The starting point is the well-known double-slit experiment as depicted in Figure 4 (left).

A particle impinges on an arrangement that contains two slits (S1 and S2), finally hits a screen, and a detector clicks if the particle impinges in a particular region of the screen. The setup involves the additional possibility of *blocking a slit*: for example, if a blockage is put behind slit S2, then the particle will either be annihilated (intuitively, this happens if it tried to travel through slit S2) or it will pass slit S1. We can now experimentally determine the probability of the detector click, conditioned on blocking (or not) one of the slits:

$$P_{12} \quad := \quad \text{Prob(click | slits } S1 \text{ and } S2 \text{ are open),}$$
$$P_1 \quad := \quad \text{Prob(click | only slit } S1 \text{ is open),}$$
$$P_2 \quad := \quad \text{Prob(click | only slit } S2 \text{ is open).}$$

If we realize an experiment of this form within classical physics (think of goal wall shooting), then we expect that $P_{12} = P_1 + P_2$. However, in QT, we find that in general

$$P_{12} \neq P_1 + P_2,$$

and this is exactly what is called *interference*. Namely, we can think of $S1$ and $S2$ as two orthonormal basis states of a two-dimensional Hilbert space that describes "which slit" the particles passes. At the time of passage, we have a state $|\psi\rangle = \alpha|S1\rangle + \beta|S2\rangle$ (more generally, a density matrix $\rho$ which could be $|\psi\rangle\langle\psi|$ but could also be a mixed state). Putting a blockage such that only slit $Si$ is open implements the transformation $\rho \mapsto \hat{P}_i \rho \hat{P}_i$, where $\hat{P}_i = |Si\rangle\langle Si|$ (if both slots are open this is $\hat{P}_{12} = \hat{P}_1 + \hat{P}_2 = \mathbf{1}$). The final detector click corresponds to a measurement operator (POVM element) $Q$, such that the probabilities are given by $P_I = \text{tr}(\hat{P}_I \rho \hat{P}_I Q)$, with $I \in \{1, 2, 12\}$. Then $P_{12} \neq P_1 + P_2$ is a consequence of the off-diagonal terms (coherences) $\langle S_1|\rho|S_2\rangle$.

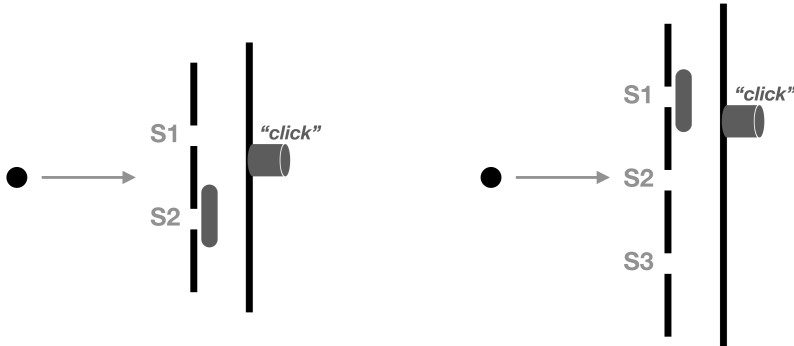

Figure 4: Interference at an $M$-slit arrangement, where $M = 2$ (left, the double-slit) resp. $M = 3$ (right, the triple-slit). For a fixed initial state preparation, we can ask for the probability of the detector click, depending on which (if any) slits are blocked. On the left, slit $S2$ is blocked, and the resulting click probability is denoted $P_1$. On the right, slit $S1$ is blocked, and the resulting click probability is denoted $P_{23}$. For the double slit, $P_{12} \neq P_1 + P_2$ is an expression of interference of order $n = 2$. For the triple-slit, QT predicts Eq. (2). A violation of this would be a novel physical phenomenon beyond QT, namely *third-order interference*.

Let us now consider a slightly more involved situation: let us add a third slit to the arrangement, as in Figure 4 (right). As before, we can block one of the slits, but now we can also block two of the slits at the same time. This allows us to define the probabilities $P_{123}, P_{12}, P_{13}, P_{23}, P_1, P_2, P_3$ in an obvious way. For example, $P_{13}$ denotes the probability of the detector click, given that (only) slit $S2$ is blocked. What we now find is that the following

identity holds according to QT:

$$P_{123} = P_{12} + P_{13} + P_{23} - P_1 - P_2 - P_3. \tag{2}$$

Why is that the case? First, note that Eq. (2) holds in *classical* physics: there, we can decompose all terms into single-slit contributions (i.e. $P_{123} = P_1 + P_2 + P_3$, $P_{12} = P_1 + P_2$ etc.). The reason why this equation also holds in the quantum case can be demonstrated as follows. Think of the initial "which-slit" state as a $3 \times 3$ density matrix $\rho = \rho_{123}$. Then we have

$$\begin{pmatrix} \bullet & \bullet & \bullet \\ \bullet & \bullet & \bullet \\ \bullet & \bullet & \bullet \end{pmatrix} = \begin{pmatrix} \bullet & \bullet & 0 \\ \bullet & \bullet & 0 \\ 0 & 0 & 0 \end{pmatrix} + \begin{pmatrix} \bullet & 0 & \bullet \\ 0 & 0 & 0 \\ \bullet & 0 & \bullet \end{pmatrix} + \begin{pmatrix} 0 & 0 & 0 \\ 0 & \bullet & \bullet \\ 0 & \bullet & \bullet \end{pmatrix}$$
$$- \begin{pmatrix} \bullet & 0 & 0 \\ 0 & 0 & 0 \\ 0 & 0 & 0 \end{pmatrix} - \begin{pmatrix} 0 & 0 & 0 \\ 0 & \bullet & 0 \\ 0 & 0 & 0 \end{pmatrix} - \begin{pmatrix} 0 & 0 & 0 \\ 0 & 0 & 0 \\ 0 & 0 & \bullet \end{pmatrix}.$$

That is, $\rho_{123} = \rho_{12} + \rho_{13} + \rho_{23} - \rho_1 - \rho_2 - \rho_3$, where $\rho_I = \hat{P}_I \rho_{123} \hat{P}_I$ for $I \in \{1, 2, 3, 12, 13, 23, 123\}$, and the projectors $\hat{P}_I$ are defined analogously to above.

In principle, however, we can imagine that nature produces an interference pattern that *violates* Eq. (2) — in this case, we would say that nature exhibits *third-order interference*.

The scheme above also gives a nice illustration why classical physics (or, rather, classical probability theory) does not admit second-order interference: namely, classical states are probability vectors, and so, for example,

$$\begin{pmatrix} \bullet \\ \bullet \\ \bullet \end{pmatrix} = \begin{pmatrix} \bullet \\ 0 \\ 0 \end{pmatrix} + \begin{pmatrix} 0 \\ \bullet \\ 0 \end{pmatrix} + \begin{pmatrix} 0 \\ 0 \\ \bullet \end{pmatrix}.$$

Thus, classically, $P_{123} = P_1 + P_2 + P_3$ (or, for two slits, $P_{12} = P_1 + P_2$).

From this starting point emerges an obvious idea: could there be physics in which the states are not tensors with one component (classical probability theory) or two (QT), but three or more? Instead of density matrices, could there be a regime of physics that is governed by "density tensors"? This idea has first appeared in the work of Hardy (2001) [40] and Wootters (1986) [89]. Recent work by Dakić *et al.* (2014) [28] constructs possible "density cube" states, but does not give a well-defined theory or state space that contains them. However, consistent theories that predict higher-order interference *can* be constructed within the framework of generalized probabilistic theories (GPTs) that we will describe next (Ududec *et al.* 2010 [83]), and the absence of third-order interference can be used as an axiom to single out QT (Barnum *et al.* 2014 [9]; see however also Barnum and Hilgert 2019 [8]). Intuitively, while CPT can arise from QT via decoherence, one might imagine that QT can similarly arise via some decoherence process from such a more general theory. However, as Lee and Selby (2018) [54] have shown, any suitable causal "super-quantum" GPT of this kind must necessarily violate the so-called *purification principle*.

The absence of third- or higher-order interference can in principle be tested experimentally, and this has in fact be done by several different groups. To the best of our knowledge, the first experimental search for third-order interference (making single photons impinge on actual spatial slit arrangements) has been performed by Sinha *et al.* (2010) [78], with negative result as expected. The relative weight of third-order contributions to the interference pattern (which is predicted to be exactly zero by QT) has been bounded to be less than $10^{-3}$ by Kauten *et al.* (2017) [50]. For other experimental approaches, see the references in this paper.

When we compare potential beyond-quantum interference with potential beyond-quantum nonlocality, then the former has an additional problem of experimental testing that the latter

does not have: to certify stronger-than-quantum correlations (if they exist), all that we have to do in principle is to design an experiment in which the structure of spacetime enforces the causal structure of a Bell scenario as depicted in Figure 2. We know how to close all potential statistical loopholes (see the recent loophole-free Bell tests, for example Giustina *et al.* 2015 [38]) to certify that an (unlikely) violation of, say, the Tsirelson bound would unambiguously falsify QT. The absence of third-order interference in the form of Eq. (2), on the other hand, makes a couple of additional assumptions. In particular, we need to ensure that the different experimental alternatives (corresponding to the different slits) are physically implemented by operations that correspond to *orthogonal projectors*.

If this is not ensured, then deviations from Eq. (2) can be detected even within standard quantum mechanics (Rengaraj *et al.* 2018 [72]). In other words: we need a physical certificate (without assuming the validity of QT) which, under the additional assumption that QT is valid, implies that the blocking transformations correspond to orthogonal projectors. This insight underlies the necessity to have a well-defined mathematical framework of *probabilistic theories* that provides a formalism to describe phenomena like higher-order interference and that tells us, for example, what the analogue (if it exists) of an orthogonal projection would be in generalizations of QT (for approaches to this particular question, see e.g. Kleinmann 2014 [51], Chiribella and Yuan 2014 [21], Chiribella and Yuan 2016 [22], Chiribella *et al.* 2020 [23]). We will describe one such framework in the next section.

# 3 Generalized probabilistic theories

The framework of generalized probabilistic theories (GPTs) has been discovered and reinvented many times over the decades, in slightly different versions (see "Further reading" below). The exposition below follows Hardy 2001 [40] and Barrett 2007 [11] (both excellent alternative references for an introduction to the GPT framework). However, the mathematical formalization will mostly follow the notation by Barnum (for example Barnum et al. 2014 [9]).

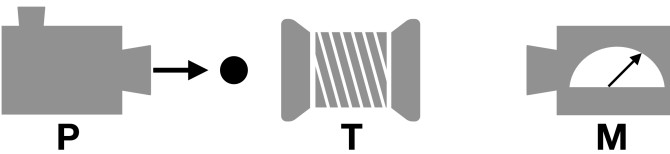

Figure 5: The paradigmatic laboratory situation that can be used to motivate the mathematical framework of GPTs: Preparation **P**, transformation **T**, and measurement **M**.

Consider the simple laboratory situation sketched in Figure 5. On every run of the experiment, a preparation device **P** spits out some physical system. In the end, a measurement device **M** will be applied to the physical system, yielding one of, for simplicity, finitely many outcomes $1, 2, \ldots, m$. In between, we may decide to apply a transformation device **T** (which may well be "do nothing"). We assume that it is meaningful to speak of the *probability* of an outcome $a$, given a choice of preparation and transformation devices: that is, we are interested in the probabilities

$$P(a|\mathbf{P}, \mathbf{T}, \mathbf{M}). \tag{3}$$

This probability can be understood in different ways: for example, we can imagine that the experiment is repeated a large number of times, yielding different outcomes on every run, with a limiting relative frequency given by $P$. Alternatively, we might be interested in doing the

experiment only once, and would like to place bets on the possible outcomes before performing it (as in a Bayesian reading of probability). Regardless of the interpretation of probability, what a GPT does is to tell us *how to describe all possible preparations, transformations and measurements of a physical system, and how to compute the outcome probabilities (3)*.

The following subsections will give an introduction to the formalism of GPTs. While I will leave out many subtleties for reasons of brevity, readers who are interested in further details are referred to the book by Holevo (2010) [45].

## 3.1 States, transformations, measurements

Since a GPT is only targeted at describing the probabilities (3) and nothing else, we have to start by removing all redundancy from the description. Consider two preparation devices **P** and **P**′. Suppose that these devices prepare a particle in exactly the same way, with the only difference that **P** was manufactured by a company called Smith&Sons, while **P**′ was manufactured by Miller&Sons. The devices have small labels with the names of the manufacturers at their bottom. Other than that, whatever we decide to measure on the prepared particles gives the exact same probabilities in both cases. Then, we should really consider **P** and **P**′ as "the same" for our purpose.

More generally, we will say that two preparation procedures **P** and **P**′ are *equivalent* if *for all possible transformations and measurements that we can in principle perform, all outcome probabilities will be identical*. In fact, we can regard every transformation **T** followed by a measurement **M** as a combined measurement **M**′. So equivalence of **P** and **P**′ can be defined by saying that *all possible measurements give identical outcome probabilities*, without specifically mentioning transformations.

Once we have introduced this equivalence relation, we can define the notion of a **state**: *a state is an equivalence class of preparation procedures*. In other words, a state subsumes all possible measurement statistics of a physical system, and nothing else.

As a more interesting example of equivalence of preparations, consider the following two procedures in quantum theory:

(i) Following a coin toss, an electron is prepared either in state $|\uparrow\rangle$ or $|\downarrow\rangle$ with probability 50% each.

(ii) The entangled state $(|\uparrow\uparrow\rangle + |\downarrow\downarrow\rangle)/\sqrt{2}$ of two electrons is prepared, and then one of the two electrons is discarded.

Both procedures amount to the preparation of the maximally mixed state $\rho = \frac{1}{2}\mathbf{1}$. In particular, spin measurements in *any* direction on the resulting physical systems will always yield completely random outcomes.

Once we have defined the notion of a state, we can also speak about the **state space** of a physical system: this is simply the collection of all possible states that can in principle be prepared by suitable preparation procedures. We will denote states by the Greek letter $\omega$, and state spaces by $\Omega$ (more details below).

Given that we aim at describing the probabilities of events, state spaces come with an important additional piece of structure: *convexity*. That is, we can always think of the following situation: a random number $i \in \{1, 2, \ldots, n\}$ is obtained with probability $p_i$ (for example via some measurement on some state, or by tossing coins), and then state $\omega_i \in \Omega$ is prepared, while $i$ is discarded. The resulting procedure will still correspond to the preparation of a physical system that leads to well-defined measurement probabilities. Hence there will be an associated state $\omega \in \Omega$. By construction, it satisfies

$$P(a|\omega, \mathbf{M}) = \sum_{i=1}^{n} p_i P(a|\omega_i, \mathbf{M}) \tag{4}$$

for all possible outcomes $a$ of all possible measurements **M**. This equation allows us to introduce a natural convex-linear structure on the state space. That is, we can write

$$\omega = \sum_{i=1}^{n} p_i \omega_i, \tag{5}$$

and by doing so introduce the useful convention that states are *elements of some vector space A over the real numbers* $\mathbb{R}$. (I am skipping several details of argumentation at this point; interested readers are again invited to look into Holevo's book). In the following, we will always assume for mathematical simplicity that this vector space is finite-dimensional. We will also denote physical systems by upper-case letters like $A$ (for example, the spin degree of freedom of an electron), the corresponding state spaces by $\Omega_A$, and the vector space on which $\Omega_A$ lives will also be denoted $A$.

Before giving some examples, let us make two more physically motivated assumptions that significantly simplify the mathematical description. Namely, let us assume that $\Omega_A$ is **compact**, i.e. topologically closed and bounded. Intuitively, $\Omega_A$ should be bounded since probabilities are bounded between zero and one, and probabilities are all we are ever computing from a state. Furthermore, suppose that we have a sequence of states $\omega_1, \omega_2, \omega_3, \dots$ that are all elements of $\Omega_A$, i.e. that can be in principle prepared on our physical system $A$. Suppose that $\lim_{n\to\infty} \omega_n = \omega$ for some element of the vector space $\omega \in A$. Is $\omega$ also a valid state? Physically, it should be: after all, we can prepare *arbitrarily good approximations to* $\omega$, and this is all we can ever hope to achieve in the laboratory anyway. This motivates to demand that $\omega \in \Omega_A$, i.e. that $\Omega_A$ is a closed set.

Furthermore, Eq. (5) above implies that *convex combinations of valid states are again valid states*, or, in other words that **state spaces are convex sets**.

This gives us (almost) the following definition:

**Definition 5.** *A state space is a pair* $(A, \Omega_A)$, *where $A$ is a real finite-dimensional vector space, and* $\Omega_A \subset A$ *is a compact convex set of dimension* $\dim \Omega_A = \dim A - 1$ *such that there is a linear "normalization functional"* $u_A : A \to \mathbb{R}$ *with* $u_A(\omega) = 1$ *for all* $\omega \in \Omega_A$. *The* state cone *is* $A_+ := \{\lambda \omega \mid \lambda \geq 0, \omega \in \Omega_A\}$.

This definition says a couple of things. First, we want the ability to mathematically describe the *normalization* of a state: basically, $u_A(\omega)$ is the probability that the physical system is there. If $\omega \in \Omega_A$, i.e. if $\omega$ is a normalized state (as "usual"), then this probability is one. However, it is often useful to talk about unnormalized states — in particular, *subnormalized* states for which $u_A(\omega') < 1$. These could come, for example, from preparing a normalized state $\omega$ and then discarding the system with probability $1 - \lambda$, resulting in the subnormalized state $\omega' = \lambda \omega$. The set $A_+$ is exactly the set of elements that can be obtained in this way, via any non-negative scalar factor $\lambda \geq 0$.

The dimension condition can then be interpreted as saying that *we choose the vector space A as small as possible*: it contains enough dimensions to carry all normalized and unnormalized states, and not more. We also see that $\Omega_A = \{\omega \in A_+ \mid u_A(\omega) = 1\}$, and $\dim A_+ = \dim A$. Furthermore, $A_+$ is a *pointed convex cone* in the sense of convex geometry (Aliprantis and Tourky, 2007 [2]): $A_+ + A_+ \subseteq A_+$, $\lambda A_+ \subseteq A_+$ for all $\lambda \geq 0$, and $A_+ \cap (-A_+) = \{0\}$.

Before turning to some examples, we introduce some further terminology:

**Definition 6.** *A state* $\omega \in \Omega_A$ *is called* pure *if it is an extremal point of the convex set* $\Omega_A$, *and otherwise* mixed.

This definition uses a basic notion of convex geometry (Webster, 1994 [86]): an *extremal point* of a convex set $\Omega$ is an element of that set that cannot be written as a non-trivial convex

combination of other points. Hence, a state $\omega \in \Omega_A$ is pure if and only if

$$\omega = \lambda \omega_1 + (1-\lambda)\omega_2 \text{ with } 0 \leq \lambda \leq 1, \omega_1, \omega_2 \in \Omega_A \;\Rightarrow\; \lambda \in \{0,1\} \text{ or } \omega_1 = \omega_2.$$

That is, if we write $\omega$ as a convex combination of $\omega_1$ and $\omega_2$, then this convex combination must be trivial: either $\omega_1 = \omega_2 \ (= \omega)$, or $\lambda$ is zero or one.

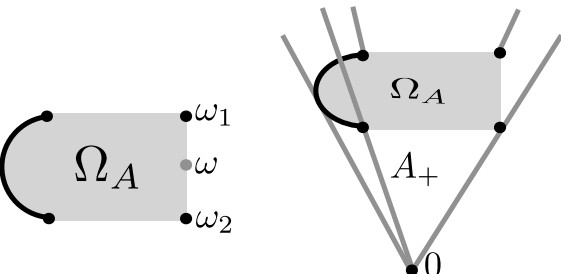

Figure 6: An arbitrary state space and its pure states (black points in $\Omega_A$).

Figure 6 gives a (somewhat arbitrary) example of a state space and of its pure states. In this example, we have the vector space $A = \mathbb{R}^3$ that carries a state cone $A_+$ (right); the set of normalized states $\Omega_A$ is depicted on the left. The pure states are those marked in black: this includes the half-circle boundary and the pure states $\omega_1$ and $\omega_2$. The state $\omega$ is on the boundary of the state space, but it is *not* pure: it can be written as the non-trivial convex combination $\omega = \frac{1}{2}\omega_1 + \frac{1}{2}\omega_2$. All states in the interior of $\Omega_A$ are mixed states, too.

The two most important examples are given by classical probability theory (CPT) and quantum theory (QT):

**Example 7** (*N*-outcome classical probability theory). *The vector space is $A = \mathbb{R}^N$, and the normalized states are the discrete probability distributions:*

$$\Omega_A = \left\{ (p_1, \ldots, p_N) \in \mathbb{R}^N \mid p_i \geq 0, \ \sum_{i=1}^{N} p_i = 1 \right\}.$$

*Geometrically, this set is a simplex. The cases $N = 2$ and $N = 3$ are depicted in Figure 7. The $N = 4$ case would correspond to a tetrahedron embedded in $\mathbb{R}^4$.*

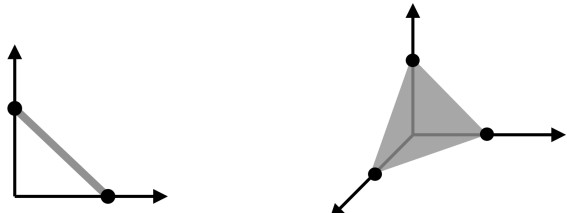

Figure 7: State spaces of classical probability (i.e. $\Omega_A$ is depicted in gray). Left: the classical bit with $N = 2$ outcomes. Right: The classical "trit" with $N = 3$.

*As one can see, the normalization functional is $u_A(p) = p_1 + p_2 + \ldots + p_N$ (where $p = (p_1, \ldots, p_N)$). The state cone $A_+$ consists of all those vectors in $\mathbb{R}^N$ that have only non-negative entries, i.e. the positive orthant. It is also not difficult to see that there are $N$ pure states, namely $p^{(i)} = (0, \ldots, 0, \underbrace{1}_{i}, 0, \ldots, 0)$ (with $i = 1, \ldots, N$). These are exactly the* deterministic *distributions.*

Convex geometry allows us to draw some conclusions about states that are valid for every GPT. For example, due to the Minkowski-Carathéodory theorem (Webster, 1994 [86]), every state $\omega \in \Omega_A$ can be written as a convex combination of at most $\dim A$ pure states. In the classical case discussed above, the corresponding decomposition of $\omega$ into pure states is unique; but in general, this is not the case, as the example of quantum theory demonstrates. To state it, we will use the notation $\mathbf{M}_n(\mathbb{C})$ for the $n \times n$ complex matrices, and $\mathbf{H}_n(\mathbb{C})$ for the *Hermitian* complex $n \times n$ matrices, i.e. those $M \in \mathbf{M}_n(\mathbb{C})$ with $M^\dagger = M$.

**Example 8** (*N*-outcome quantum theory). *The vector space is $A = \mathbf{H}_N(\mathbb{C})$ — note that this is a vector space over the* reals*, not over the complex numbers, since it is not closed under multiplication with the imaginary unit i. The set of normalized states is the set of* density matrices*,*

$$\Omega_A = \{\rho \in A \mid \rho \geq 0, \ \mathrm{tr}(\rho) = 1\}.$$

*Here and in the following, the notation $\rho \geq 0$ denotes the fact that $\rho$ is positive-semidefinite, i.e. $\langle\psi|\rho|\psi\rangle \geq 0$ for all $|\psi\rangle \in \mathbb{C}^N$. This is equivalent to $\rho$ being Hermitian and having only non-negative eigenvalues.*

*The normalization functional is $u_A(\rho) := \mathrm{tr}(\rho)$, and the state cone becomes the* positive semidefinite cone, $A_+ = \{M \in A \mid M \geq 0\}$.

In the quantum case, the general definition of a "pure state" (Definition 6 above) reduces to the usual definition of a pure quantum state: every density matrix can be diagonalized, $\rho = \sum_{i=1}^{N} p_i |i\rangle\langle i|$ for some orthonormal basis $\{|i\rangle\}_{i=1}^{N}$, and if the $p_i$ are not all zero or one, then this defines a non-trivial convex decomposition of $\rho$ into other states. Hence $\rho$ is pure if and only if it can be written as a one-dimensional projector, i.e. as $\rho = |\psi\rangle\langle\psi|$ for some suitable $|\psi\rangle \in \mathbb{C}^N$.

What do the quantum state spaces look like – geometrically, as convex sets? For the case $N = 2$ (the qubit), the answer is simple and easy to depict (see e.g. Nielsen and Chuang, 2000 [63]): we can write every $2 \times 2$ density matrix $\rho$ in the form

$$\rho = \frac{1}{2}\begin{pmatrix} 1 + r_3 & r_1 - ir_2 \\ r_1 + ir_2 & 1 - r_3 \end{pmatrix}.$$

With the "Bloch vector" $r = (r_1, r_2, r_3)$, we have the equivalence

$$\rho \geq 0 \iff \lambda_{1/2} = \frac{1}{2}\left(1 \pm \sqrt{r_1^2 + r_2^2 + r_3^2}\right) = \frac{1}{2}(1 \pm |r|) \geq 0,$$

where $\lambda_{1/2}$ denotes the two eigenvalues of $\rho$. This parametrization identifies the qubit state space $\Omega_A$ with the *Bloch ball* – the unit ball in three dimensions. It is crucial that this parametrization is *linear*, so that we can interpret convex mixtures in the ball as probabilistic mixtures of states. The Bloch ball is sketched in Figure 8.

As expected, the pure states lie in the boundary of the state space – but in this case, *every boundary point* is in fact a pure state. This is a very special property (called "strict convexity" of $\Omega_A$) that is generically absent, as Figure 6 shows. Note that this property also holds for the classical bit (a one-dimensional line segment).

What about $N \geq 3$ — are these state spaces also balls, perhaps of some higher dimension? A moment's thought shows that this cannot be the case: all these state spaces contain mixed states in their topological boundary. For example, for $N = 3$, the state $\rho = \mathrm{diag}\left(\frac{1}{2}, \frac{1}{2}, 0\right)$ is mixed but lies on the boundary of the set of density matrices (there are unit trace matrices with negative eigenvalues arbitrarily close to that state). Hence these state spaces cannot be strictly convex, and in particular, they cannot correspond to Euclidean balls. Instead, these state spaces are compact convex sets with quite complex and intriguing structure. A beautiful

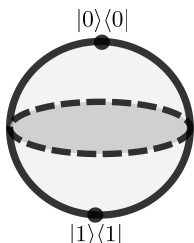

Figure 8: The state space of a quantum bit can be represented as a three-dimensional ball, the Bloch ball. The pure quantum states lie on the boundary (the sphere), with orthogonal states on antipodal points of the sphere. The center represents the mixed state $\frac{1}{2}\mathbf{1}$. In contrast to the classical states, convex decompositions of mixed states into pure states are not unique.

attempt to visualize the $N = 3$ case can be found in Bengtsson *et al.*, 2013 [14], and a much more comprehensive introduction to the geometry of quantum states is given in the book by Bentsson and Życzkowski, 2017 [15].

So far, we have only talked about *states*; let us now see how to describe *measurements* in the GPT framework. Our starting point are Eq. (4) and (5), which say that probabilistic mixtures of preparation procedures should lead to the identical mixtures of the corresponding outcome probabilities. In other words, *outcome probabilities of measurements must be linear functionals of the state*. This motivates the following definition.

**Definition 9** (Measurements)**.** *Let $(A, \Omega_A)$ be a state space. A linear functional, i.e. an element of the dual space $e \in A^*$, is an* effect *if it attains values between zero and one on all normalized states, i.e. $0 \le e(\omega) \le 1$ for all $\omega \in \Omega_A$.*

*An n-outcome measurement (with $n \in \mathbb{N}$ arbitrary) is a collection of effects $e^{(1)}, \ldots, e^{(n)}$ with the property that $e^{(1)} + \ldots + e^{(n)} = u_A$.*

*The* effect cone *is $A_+^* = \{\lambda \cdot e \mid e \text{ is an effect, } \lambda \ge 0\}$.*

If we prepare a system in state $\omega \in \Omega_A$ and perform an *n*-outcome measurement, then the probability of the *i*-th outcome must certainly lie in the interval $[0, 1]$, and it must be a linear functional of the state: hence it must be given by $e^{(i)}(\omega)$, where $e^{(i)}$ is some effect. The total outcome probability must be unity, so that $e^{(1)}(\omega) + \ldots + e^{(n)}(\omega) = 1$ for all $\omega \in \Omega_A$. Since the normalized states span the vector space $A$, this is only possible if $e^{(1)} + \ldots + e^{(n)} = u_A$, the normalization functional.

The effect cone is an object of mathematical convenience. In convex geometry terminology (Aliprantis and Tourky, 2007 [2]), this is exactly the *dual cone* of $A_+$, i.e.

$$A_+^* = \{e \in A^* \mid e(\omega) \ge 0 \text{ for all } \omega \in A_+\}.$$

Sometimes, we will call the elements of $A_+^*$ "unnormalized effects" (since their value can be larger than one on some states). There are a couple of interesting properties of the dual cone; for example, in our case, $A_+^{**} = A_+$. In other words, the unnormalized effects are exactly the functionals that give non-negative values on all unnormalized states – and *the unnormalized states are exactly the vectors that give non-negative values on all unnormalized effects*. This expresses a certain form of duality between states and effects.

Let us now discuss the **measurements in CPT and QT**. In $N$-outcome CPT, the vector space is $A = \mathbb{R}^N$, and it is convenient for us to identify it with its dual space via the dot product, $A \simeq A^*$. Effects are then vectors too, such that $e(\omega) = e \cdot \omega$. To check that a functional $e = (e_1, \ldots, e_N)$ is a valid effect, it is sufficient to check that it yields probabilities in the interval $[0, 1]$ on all

*pure* states (the rest follows from convexity). Clearly, this is true if and only if $0 \leq e_i \leq 1$ for all $i$ — i.e. *e is a valid effect if all its entries lie in the unit interval*. In particular, the normalization functional is a valid effect (as always), and $u_A = (1, 1, \ldots, 1)$. The effect cone $A_+^*$ is thus the set of all vectors with non-negative entries — which is the same as the state cone.

For $N$-outcome QT, let us also identify $A = \mathbf{H}_N(\mathbb{C})$ and its dual space via some inner product, namely $\langle X, Y \rangle := \mathrm{tr}(XY)$ the Hilbert-Schmidt inner product. For example, this means that the normalization functional becomes the unit matrix, $u_A = \mathbf{1}$, since $u_A(\rho) = \mathrm{tr}(\rho) = \langle \mathbf{1}, \rho \rangle$. An effect is now a self-adjoint matrix $E$ with $0 \leq \mathrm{tr}(E\rho) \leq 1$. As in the CPT case, it is sufficient to check this on pure states $\rho = |\psi\rangle\langle\psi|$, such that $0 \leq \langle\psi|E|\psi\rangle \leq 1$ for all normalized $\psi \in \mathbb{C}^N$. But this is equivalent to $0 \leq E \leq \mathbf{1}$ — that is, both $E$ and $\mathbf{1} - E$ must be positive-semidefinite. A collection of $n$ such effects, $E_1, E_2, \ldots, E_n$ with $E_1 + E_2 + \ldots + E_n = \mathbf{1}$, while each $E_i$ is positive-semidefinite, is known as a *POVM*: a positive operator-valued measure. Indeed, the set of all possible quantum measurements are given by POVMs, and we have just rederived this fact within the framework of GPTs.

Therefore, in this case, we also obtain that $A_+^* = A_+$: the effect and state cones coincide. One might be tempted to conjecture that this is true in general, but it is easy to see that it is not. This property of **strong self-duality** – that there exists some inner product such that the state and effect cones are identical – is a remarkable property that holds only under very special conditions. It is known to be false, for example, for the case that $\Omega_A$ is a square (sometimes called a "gbit"; Barrett 2007 [11]), and it is also false for the example in Figure 6. However, self-duality is known to follow from a strong symmetry property called *bit symmetry*. Call every pair of pure and perfectly distinguishable states $\omega_1, \omega_2$ a *bit*. Suppose that for every pair of bits $(\omega_1, \omega_2)$ and $(\varphi_1, \varphi_2)$, there is a reversible transformation $T$ such that $T\omega_1 = \varphi_1$ and $T\omega_2 = \varphi_2$ (this is true for CPT and QT, for example). Then strong self-duality follows (Müller and Ududec, 2012 [60]).

In summary, while states and measurements are described by the same kinds of objects (positive semidefinite matrices) in QT, they will be described by different sets of objects in general GPTs. Moreover, measurements in GPTs can have properties that look quite unusual from the perspective of QT. Consider the following definition:

**Definition 10.** *Let $(A, \Omega_A)$ be some state space. A set of states $\omega_1, \ldots, \omega_n$ is called (jointly) perfectly distinguishable if there exists a measurement $e^{(1)}, \ldots, e^{(n)}$ with $e^{(i)}(\omega_j) = \delta_{ij}$ (that is 1 if $i = j$ and 0 otherwise).*

*The maximal number $n \in \mathbb{N}$ for which there exists a set of $n$ perfectly distinguishable states is called the* capacity *of the state space, and is denoted $N_A$. On the other hand, we will denote the dimension of the state space by $K_A := \dim A$.*

In other words, $n$ states are perfectly distinguishable if we can in principle build a detector that, on feeding it with one of the states, tells us with certainty which of the states it was that we have initially prepared (assuming that we are promised that we have indeed prepared one of the states and not another one).

In QT, the $\omega_i$ are density matrices, and they are perfectly distinguishable if and only if their supports are mutually orthogonal (if all states are pure, $\omega_i = |\psi_i\rangle\langle\psi_i|$, this means that $\langle\psi_i|\psi_j\rangle = \delta_{ij}$). The capacity $N_A$ in QT is hence equal to the dimension of the underlying Hilbert space. The dimension of the state space, $K_A$, is the number of real parameters in a Hermitian $N_A \times N_A$-matrix. Simple parameter counting shows that this is $K_A = N_A^2$. In particular, the set of normalized states has dimension $\dim \Omega_A = N_A^2 - 1$, which equals three for the qubit, in accordance with Figure 8.

In CPT, on the other hand, we have $K_A = N_A$, which is equal to the cardinality of the sample space on which the states are defined as probability distributions.

In particular, both in CPT and in QT, *if some states are pairwise perfectly distinguishable, then they are automatically jointly perfectly distinguishable*. After all, the condition of joint

distinguishability is pairwise orthogonality of the supports. It is perhaps surprising to see that this statement is in general false for GPTs. Let us illustrate this with an example.

**Example 11** (The gbit (Barrett 2007 [11])). *The generalized bit, or "gbit", is a state space* $(A, \Omega_A)$*, where* $A = \mathbb{R}^3$*, and*

$$\Omega_A = \left\{ (x, y, 1) \in \mathbb{R}^3 \mid -1 \leq x \leq 1, \; -1 \leq y \leq 1 \right\}.$$

*In particular, the normalization functional is* $u_A(x, y, z) := z$*. If we identify A with its dual space via the dot product, then* $u_A = (0, 0, 1)$*. This state space has four pure states (the corners of the square):*

$$\omega_1 = (-1, -1, 1), \quad \omega_2 = (-1, 1, 1), \quad \omega_3 = (1, 1, 1), \quad \omega_4 = (1, -1, 1).$$

*It is a simple exercise to work out the set of effects and the effect cone. Writing* $\omega = (x, y, z)$ *and demanding that* $e(\omega) \in [0, 1]$ *for all* $\omega \in \Omega_A$*, we find in particular the following effects:* $e^{(x)}(\omega) = \frac{1}{2}(z + x)$*,* $\bar{e}^{(x)}(\omega) = \frac{1}{2}(z - x)$*,* $e^{(y)}(\omega) = \frac{1}{2}(z + y)$*,* $\bar{e}^{(y)}(\omega) = \frac{1}{2}(z - y)$*. Writing these as vectors, we get*

$$e^{(x)} = \frac{1}{2}(1, 0, 1), \quad \bar{e}^{(x)} = \frac{1}{2}(-1, 0, 1), \quad e^{(y)} = \frac{1}{2}(0, 1, 1), \quad \bar{e}^{(y)} = \frac{1}{2}(0, -1, 1).$$

*It turns out that the set of effects is the convex hull of these effects, the "unit effect" (normalization functional)* $u_A$*, and the zero effect* 0*. Geometrically, this can be depicted as in Figure 9. Note that* $A_+$ *and* $A_+^*$ *are not identical. Even if we had chosen another inner product (rather than the dot product) to represent the effects, then the two cones would never align. This is because the gbit is not strongly self-dual (Janotta* et al.*, 2011 [46]).*

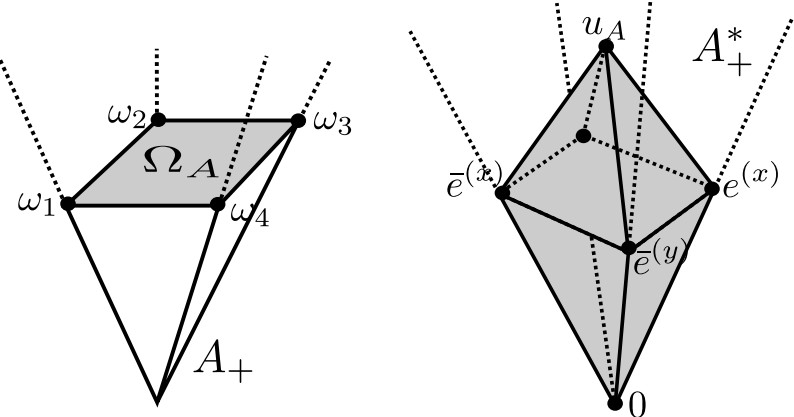

Figure 9: The "gbit" state space (generalized bit). Left: the state cone, with the normalized states in gray. Right: the effect cone, with the set of effects in gray.

*More importantly, we now see that every two distinct pure states are perfectly distinguishable. For example, consider the pure states* $\omega_1$ *and* $\omega_2$*, and consider the two effects* $e^{(y)}$ *and* $\bar{e}^{(y)}$*. Note that these two effects constitute a valid measurement:* $e^{(y)} + \bar{e}^{(y)} = u_A$*. Moreover, we have* $e^{(y)}(\omega_1) = 0$ *and* $e^{(y)}(\omega_2) = 1$*, and hence* $\bar{e}^{(y)}(\omega_1) = 1$ *and* $\bar{e}^{(y)}(\omega_2) = 0$*. That is, the measurement* $(e^{(y)}, \bar{e}^{(y)})$ *perfectly distinguishes the states* $(\omega_1, \omega_2)$*.*

*A similar construction can be made for all other pairs of pure states. However, one can check that no three states of* $\Omega_A$ *are jointly perfectly distinguishable. On the one hand, this shows that the capacity of this state space is* $N_A = 2$*, hence the name "gbit" (and not gtrit etc.); on the other hand, it demonstrates that pairwise perfect distinguishability does not in general imply joint perfect distinguishability.*

The final ingredient of Figure 5 that we have not yet discussed so far are the **transformations**. Clearly, a transformation $T$ maps an ingoing state $\omega$ to some outgoing state $\omega' = T\omega$. In general, we can think of transformations from one physical system to another one (for example, mapping the spin state of an electron to the state of a quantum dot, or maps between Hilbert spaces of different dimensions), but for simplicity, let us focus on transformations for which in- and outgoing systems are the same. Consider the following two scenarios, where $\lambda_1, \ldots, \lambda_n$ are probabilities summing to unity, and $\omega_1, \ldots, \omega_n$ are normalized states:

(i) A preparation device prepares the state $\omega_i$ with probability $\lambda_i$, resulting in the mixed state $\omega = \sum_i \lambda_i \omega_i$. This mixed state is sent into the transformation device, resulting in some final state $\omega' = T\omega$.

(ii) With probability $\lambda_i$, a preparation device prepares the state $\omega_i$, which is then sent into the transformation device, resulting in the final state $\omega_i' = T\omega_i$.

Clearly, (i) and (ii) are different descriptions of one and the same laboratory procedure; they must hence result in the exact same statistics of any measurements that we may decide to perform in the end, and therefore lead to the same final state $\omega'$. But this implies that $T\left(\sum_{i=1}^n \lambda_i \omega_i\right) = \sum_{i=1}^n \lambda_i T(\omega_i)$: transformations must be linear. This motivates the following definition.

**Definition 12** (Transformation). *Let $(A, \Omega_A)$ be some state space. A* transformation *is a linear map $T : A \to A$ with $T(\Omega_A) \subseteq \Omega_A$, i.e. every normalized state is mapped to another normalized state. A transformation $T$ is* reversible *if it is invertible as a linear map and if $T^{-1}$ is a transformation, too.*

*A* dynamical state space *is a triplet $(A, \Omega_A, \mathcal{T}_A)$, where $(A, \Omega_A)$ is a state space, and $\mathcal{T}_A$ is a compact (or finite) group of reversible transformations.*

This definition subsumes several properties of transformations that are either necessary or desirable. First, transformations $T$ must map valid input state *to valid output states*; second, they should preserve the normalization. This leads to the demand that $T(\Omega_A) \subseteq \Omega_A$. It also implies that $u_A \circ T = u_A$ for every transformation $T$: the normalization after the transformation is the same as before. In some situations, it is important to allow a larger class of *normalization-nonincreasing* transformations; for example, when we are interested in filters and projections as in the case of higher-order interference. For more details on this, see e.g. Barrett 2007 [11] or Ududec *et al.*, 2010 [83].

Of particular significance are transformations that can be physically undone after they have been implemented: these are the reversible transformations. Namely, after applying $T$ to some state, we can apply $T^{-1}$ to the resulting state, with the total effect of doing nothing. In the following, we will restrict our attention to those, because they will turn out to be particularly important in the context of axiomatic reconstructions of QT. By definition, reversible transformations satisfy $T(\Omega_A) \subseteq \Omega_A$ and $T^{-1}(\Omega_A) \subseteq \Omega_A$ — but this is only possible if $T(\Omega_A) = \Omega_A$. In other words, *every reversible transformation is a linear symmetry of the state space*.

If we write down any transformation $T$, then it satisfies all the conditions that we need for a map to be a physically implementable operation on some state — as long as the state space is considered in isolation. In many cases, however, state spaces are subsystems of larger state spaces; this is certainly the case in standard laboratory situations, where a bit or qubit is usually embedded into some sort of infinite-dimensional Hilbert space or operator algebra. Further below, we will discuss how GPT state spaces can be combined via generalizations of QT's tensor product rule to form larger state spaces. In these cases, additional conditions may arise from the demand that the transformations are also valid processes when the state spaces on which they act are part of a larger state space.

Definition 12 incorporates this insight by introducing the formal possibility that *not all mathematically valid transformations are in fact physically allowed*. Among the reversible transformations in particular, a dynamical state space comes with an additional choice of a set $\mathcal{T}_A$ of allowed reversible transformations. The only demand is that if $T$ is an allowed reversible transformation and so is $T'$, then $T' \circ T$ is too: after all, we can implement one transformation after the other. Similarly, "do nothing" should be an allowed transformation, i.e. $\mathbf{1} \in \mathcal{T}_A$; and the meaning of reversibility demands that $T^{-1} \in \mathcal{T}_A$ whenever $T \in \mathcal{T}_A$. This makes $\mathcal{T}_A$ a group. The demand that $\mathcal{T}_A$ should be topologically closed and bounded can be motivated similarly as we did in the case of the state space. Hence $\mathcal{T}_A$ is a compact matrix group.

**Example 13** (Reversible transformations in QT). *In Example 8, we have defined the N-outcome quantum state space as the set of $N \times N$ density matrices. Based on this definition, we have already derived the most general form of* measurements *in QT: these are the POVMs. Let us now use the GPT framework to deduce the most general form of* reversible transformations.

*As discussed above, a reversible transformation $T$ is an (invertible) linear map $T : \mathbf{H}_N(\mathbb{C}) \to \mathbf{H}_N(\mathbb{C})$ which is a* linear symmetry of the state space. *That is, it must map the set $\Omega_A$ of $N \times N$ density matrices onto itself, and it must be linear in the sense of preserving real-linear combinations of Hermitian matrices. (A priori this is completely unrelated to linearity on state vectors in the Hilbert space.) Determining the most general transformations $T$ with this property is hence a mathematical exercise. Its solution is known as* Wigner's theorem *(see e.g. Bargmann 1964 [4]). Namely, these turn out to be the transformations of the following form:*

$$\rho \mapsto U \rho U^{-1},$$

*where $U$ is a unitary or antiunitary map. Which of these transformations can actually be implemented physically depends on auxiliary assumptions. If we only consider single quantum state spaces in isolation, then there is no a priori reason to regard antiunitary transformations as physically impossible. However, if — as in actual physics — we consider a full theory of state spaces, combining via the usual tensor product rule, then only the unitary transformations can be implemented. This is because antiunitary maps like the transposition, $\rho \mapsto U \rho U^{-1} = \rho^\top$, are known to generate negative eigenvalues when applied to half of an entangled state. In other words, unitary transformations are* completely positive *while antiunitary transformations are not (Nielsen and Chuang 2000 [63]).*

*Thus, in our physical world, quantum systems come as dynamical state spaces $(A, \Omega_A, \mathcal{T}_A)$, where $A$ and $\Omega_A$ are as defined in Example 8, and $\mathcal{T}_A$ is the group of unitary conjugations, $\rho \mapsto U \rho U^\dagger$. This is a* subgroup *of the full group of reversible transformations.*

The argumentation so far yields a very interesting perspective on the **superposition principle** of QT. Usually, the superposition principle is seen as some kind of fundamental (and mysterious) principle or axiom of QT. In our case, however, we can see it as some kind of *accidental mathematical consequence* of the shape of the state space. It just so happens to be the case that the pure states are of the form $|\psi\rangle\langle\psi|$, and applying reversible transformations to them yields $|\psi\rangle\langle\psi| \mapsto U|\psi\rangle\langle\psi|U^\dagger$. Restricting our attention to the pure states only, we can thus simplify the mathematical description by "taking the square root" in some sense, and consider the map $|\psi\rangle \to U|\psi\rangle$ only (without forgetting that we have to disregard arbitrary phase factors).

But doesn't this argumentation defer the question of "why the superposition principle" to "why the density matrices"? At first sight it seems so, but we will see later that we can derive the shape of the state space — that is, that it must correspond to the set of density matrices — from simple information-theoretic principles. The superposition principle will then indeed follow as an accidental consequence in the way just described.

We leave it for the reader as an exercise to check that the **reversible transformations in CPT** are the permutations: $(p_1, \ldots, p_N) \mapsto (p_{\pi(1)}, \ldots, p_{\pi(n)})$, where $\pi : \{1, \ldots, n\} \to \{1, \ldots, n\}$ is one-to-one. For the gbit as defined in Example 11, those transformations are of the form $T = \begin{pmatrix} D & 0 \\ 0 & 1 \end{pmatrix} \in M_3(\mathbb{R})$, where $D \in M_2(\mathbb{R})$ is any element of $D_2$, the *dihedral group* of order 4 (the symmetry group of the square).

Above, we have admitted the possiblity that *not all mathematically well-defined transformations are in fact "physically" allowed*, but further above, our formalism has forced the state cone $A_+$ and the effect cone $A_+^*$ to be full duals of each other. In other words, we are implicitly working under the so-called **"no-restriction hypothesis"** (Chiribella, d'Ariano and Perinotti, 2010 [19]): for any given set of states, all mathematically well-defined effects can in principle be implemented (and vice versa). Here we make this assumption mainly for reasons of simplicity, but there are situations in which a more general approach is warranted, e.g. in the context of stabilizer quantum mechanics or Spekkens' toy theory (Spekkens 2007 [80]). In this case, one could define, for example, a "sub-cone" of physically allowed effects (though this is not the only possibility). For more details on such a more general approach, see e.g. Janotta and Lal, 2013 [48].

In the example of QT, we have seen that we can represent a quantum bit in two ways: either as the set of $2 \times 2$ density matrices, or as the three-dimensional Bloch ball. In fact, all probabilistic predictions are exactly identical for both descriptions. This is an example of a general freedom that we have in representing GPTs:

**Definition 14** (Equivalent state spaces). *Two state spaces $(A, \Omega_A)$ and $(B, \Omega_B)$ are called equivalent if there exists an invertible linear map $L : A \to B$ such that $\Omega_B = L(\Omega_A)$. Two dynamical state spaces $(A, \Omega_A, \mathcal{T}_A)$ and $(B, \Omega_B, \mathcal{T}_B)$ are called equivalent if they additionally satisfy $\mathcal{T}_B = L\mathcal{T}_A L^{-1}$.*

Equivalent state spaces are indistinguishable in all of their probabilistic properties: to every state $\omega_A \in \Omega_A$, there is a corresponding state $\omega_B = L\omega_A \in \Omega_B$; to every effect $e_A \in A_+^*$ there is a corresponding effect $e_B = e_A \circ L^{-1} \in B_+^*$. Finally, to every transformation $T_A \in \mathcal{T}_A$ there is a corresponding transformation $T_B = LT_A L^{-1} \in \mathcal{T}_B$, such that the outcome probabilities are the same: $e_B T_B \omega_B = e_A L^{-1} L T_A L^{-1} L \omega_A = e_A T_A \omega_A$. Intuitively, equivalent state spaces are of the same convex shape. In particular, equivalent state spaces must have the same dimensions.

**Example 15.** *As illustrated in Example 8, the Bloch ball representation and the density matrix representation of the quantum bit are equivalent. In more detail, the dynamical state spaces $(A, \Omega_A, \mathcal{T}_A)$ and $(B, \Omega_B, \mathcal{T}_B)$ are equivalent, where*

$$
A = \mathbb{R}^4, \quad \Omega_A = \left\{ \begin{pmatrix} 1 \\ r \end{pmatrix} \,\middle|\, r \in \mathbb{R}^3, \ |r| \le 1 \right\}, \quad \mathcal{T}_A = \left\{ \begin{pmatrix} 1 & 0 \\ 0 & R \end{pmatrix} \,\middle|\, R \in \mathrm{SO}(3) \right\},
$$
$$
B = \mathbf{H}_2(\mathbb{C}), \quad \Omega_B = \{\rho \in B \mid \mathrm{tr}(\rho) = 1, \ \rho \ge 0\}, \quad \mathcal{T}_B = \left\{ \rho \mapsto U\rho U^\dagger \mid U \text{ is unitary} \right\},
$$

*and an invertible linear map $L : A \to B$ that establishes this equivalence is given by*

$$
L(r_0, r_1, r_2, r_3) := \frac{1}{2} \begin{pmatrix} r_0 + r_3 & r_1 - ir_2 \\ r_1 + ir_2 & r_0 - r_3 \end{pmatrix}.
$$

In many places in the literature (for example in Barrett 2007 [11]), one finds an alternative route to the GPT framework. This alternative approach begins with the same laboratory situation as in Figure 5, but argues less abstractly. It postulates in a more concrete manner that *states are "lists of probabilities"*, corresponding to a set $e_1, \ldots, e_n$ of **"fiducial effects"** — a bunch of outcome probabilities that are *sufficient to fully characterize the state*. For example, for a qubit, these four effects might correspond to the normalization effect, and to the probabilities of measuring "spin-up" in $x$-, $y$- and $z$-directions. (Note that these effects do *not* in

general constitute a "measurement" in the sense of Definition 9, i.e. they cannot in general be jointly measured). Knowing these probabilities determines the state and hence all the outcome probabilities of all other possible measurements.

According to this prescription, a state is then simply a list of probabilities

$$(e_1(\omega), \ldots, e_n(\omega)).$$

How does this fit our definition? To see this, consider any state space $(A, \Omega_A)$ in the sense of Definition 5. Since $\Omega_A$ is a compact convex set, we can always find some invertible linear map $L : A \to A$ (in fact, many) that maps $\Omega_A$ into the unit cube $\mathcal{C} := \{(x_1, \ldots, x_n) \in A \mid 0 \leq x_i \leq 1$ for all $i\}$, where $n = \dim A$. Thus, $(A, \Omega_A)$ is equivalent to $(B, \Omega_B)$, where $B = A$ and $\Omega_B = L(\Omega_A)$. Now, every state $\omega = (\omega_1, \ldots, \omega_n) \in \Omega_B$ satisfies $e_i(\omega) := \omega_i \in [0, 1]$ by construction, for all $i \in \{1, \ldots, n\}$. Hence, in particular, the $e_i$ are valid effects, and they fit Barrett's definition of "fiducial effects".

So far, we have only considered single state spaces. In the next subsection, we will see how GPT state spaces can be combined in a way that generalizes the tensor product rule of QT.

## 3.2 Composite state spaces

Given two state spaces $(A, \Omega_A)$ and $(B, \Omega_B)$, then how can we define a meaningful composite $AB$? The philosophy of the GPT framework is not to ask for a formal rule in the first place, but to strive for the representation of fundamental operational properties that should be captured by such a formalism.

Let us therefore imagine two laboratories that are each *locally* holding systems which are described by state spaces $(A, \Omega_A)$ and $(B, \Omega_B)$. If these are two separated distinguishable laboratories, then we ought to be able to imagine that Alice performs a local experiment, and Bob *independently* performs another local experiment. For example, what Alice can do is to prepare a state $\omega_A$ and ask whether the outcome (effect) $e_A$ happens in her subsequent measurement; the probability of this is $e_A(\omega_A)$. Similarly, Bob can prepare a state $\omega_B$ and observe whether outcome $e_B$ happens, which has probability $e_B(\omega_B)$. Now we can regard this as a single joint experiment, asking whether *both* outcomes have happened. The independent joint preparations should correspond to some valid state $\omega_{AB} \in \Omega_{AB}$ of the two laboratories, and the independent joint measurement (or rather its "yes"-outcome) should correspond to a valid effect $e_{AB}$. Due to statistical independence, the joint probability must be

$$e_{AB}(\omega_{AB}) = e_A(\omega_A) \cdot e_B(\omega_B).$$

Without loss of generality, this allows us to introduce a particular piece of notation: let us write $\omega_{AB} := \omega_A \otimes \omega_B$ for the independent preparations of the two states, and $e_{AB} = e_A \otimes e_B$ for the independent measurements. Statistical mixtures (i.e. convex combinations) of states on $A$ (or on $B$) must lead to the corresponding statistical mixtures on $AB$, which tells us that $\otimes$ must be a bilinear map. Thus, reading $\otimes$ as the usual tensor product of real linear spaces, this will reproduce the correct probabilities.

What we have found at this point is that the joint vector space $AB$ that carries the composite state space must contain the tensor product space $A \otimes B$ as a subspace. This is because for every $\omega_A \in \Omega_A$ and for every $\omega_B \in \Omega_B$, we postulate that there is a state $\omega_A \otimes \omega_B \in \Omega_{AB}$ that describes the independent local preparation of the two states. This implies, on the one hand, that the convex hull of $\Omega_A \otimes \Omega_B$ is contained in $\Omega_{AB}$, and, on the other hand, that $K_{AB} \geq K_A K_B$, where we have used the notation $K_A := \dim A$ of Definition 10. But this neither tells us what the set $\Omega_{AB}$ is, nor does it tell us the vector space $AB$ or its dimension $K_{AB}$.

To narrow the possibilities down in an operationally meaningful way, let us make an additional assumption that is often (but not always) made in the GPT framework. This is a

principle called **Tomographic Locality** (Hardy, 2001 [40]): *all states $\omega_{AB} \in \Omega_{AB}$ are uniquely determined by the joint statistics of all* local *measurements.*

Fundamentally, this amounts to a claim of what we even *mean* by a joint state: the joint state is the thing that tells us all there is to know about the outcomes of local measurements and their correlations (but not more). Formally, this means the following. Take any two states $\omega_{AB}, \varphi_{AB} \in \Omega_{AB}$. If

$$e_A \otimes e_B(\omega_{AB}) = e_A \otimes e_B(\varphi_{AB})$$

for *all* local effects $e_A \in A_+^*$ and $e_B \in B_+^*$, then $\omega_{AB} = \varphi_{AB}$. In other words, *state tomography can be performed locally*, hence the name of the principle.

Due to linear algebra, this implies that $A_+^* \otimes B_+^*$ linearly spans all of the dual space $(AB)^*$ — or, in other words, that $AB = A \otimes B$. This is also equivalent to the claim that $K_{AB} = K_A K_B$.

This still does not tell us what the joint state space $\Omega_{AB}$ is — and this is in fact the generic situation in GPTs: given two state spaces, there are in general *infinitely many inequivalent possible composites* that satisfy the principle of Tomographic Locality. The full range of possibilities is captured by the following definition.

**Definition 16.** *Let $(A, \Omega_A)$ and $(B, \Omega_B)$ be state spaces. A (tomographically local) composite is a state space $(AB, \Omega_{AB})$, where $AB = A \otimes B$ and $\Omega_{AB}$ is some compact convex set satisfying*

$$\Omega_{AB}^{\min} \subseteq \Omega_{AB} \subseteq \Omega_{AB}^{\max}.$$

*The composites $(AB, \Omega_{AB}^{\min})$ and $(AB, \Omega_{AB}^{\max})$ are called the* minimal and maximal tensor products *of $(A, \Omega_A)$ and $(B, \Omega_B)$, and they are defined as follows:*

$$\begin{aligned} \Omega_{AB}^{\min} &:= \mathrm{conv}\{\omega_A \otimes \omega_B \mid \omega_A \in \Omega_A, \ \omega_B \in \Omega_B\}, \\ \Omega_{AB}^{\max} &:= \{\omega_{AB} \in AB \mid u_A \otimes u_B(\omega_{AB}) = 1, \ e_A \otimes e_B(\omega_{AB}) \geq 0 \, \forall \, e_A \in A_+^*, e_B \in B_+^*\}. \end{aligned}$$

*In the case of* dynamical *state spaces, we demand that $\mathcal{T}_A \otimes \mathcal{T}_B \subseteq \mathcal{T}_{AB}$. As a consequence, the normalization functional on $AB$ is $u_{AB} = u_A \otimes u_B$.*

In other words, $\Omega_{AB}^{\min}$ is the smallest possible composite: it only contains the product states and their convex combinations and not more. This is the necessary minimum to describe independent local state preparations. On the other hand, $\Omega_{AB}^{\max}$ is the largest possible composite: it contains all vectors that lead to non-negative probabilities on local measurements. This is the maximal possible state space that still admits the implementation of all independent local measurements. Any compact convex state space that lies "in between" these two extreme possibilities is a possible composite in the GPT framework.

**Example 17** (Composition of quantum state spaces)**.** *Consider the $M$-outcome quantum state space $(A, \Omega_A)$, and the $N$-outcome quantum state space $(B, \Omega_B)$. The usual tensor product rule of QT tells us that the composite state space should be $(AB, \Omega_{AB})$, where*

$$AB = \mathbf{H}_M(\mathbb{C}) \otimes \mathbf{H}_N(\mathbb{C}) \simeq \mathbf{H}_{MN}(\mathbb{C}), \quad \Omega_{AB} = \{\rho \in AB \mid \mathrm{tr}(\rho) = 1, \ \rho \geq 0\}.$$

*In other words, the usual tensor product rule of QT tells us that the composite is simply the $(MN)$-outcome quantum state space. This composite satisfies the principle of Tomographic Locality. This can be checked, for example, by noting that $\dim \mathbf{H}_M(\mathbb{C}) = M^2$, and thus*

$$K_{AB} = (MN)^2 = M^2 N^2 = K_A \cdot K_B.$$

*This is* strictly in between the minimal and maximal tensor products. *Namely, $\Omega_{AB}^{\min}$ corresponds to the set of states that can be written as convex combinations of product states: the* separable states. *On the other hand, $\Omega_{AB}^{\max}$ corresponds to the set of operators that yield non-negative probabilities for all local measurements, which includes operators that are not density matrices: so-called* witnesses *or POPT states (Barnum* et al.*, 2010 [6]). These operators have negative eigenvalues, but these would only be manifested on performing* entangled *measurements.*

The composition of *classical* state spaces (those of Example 7) satisfy Tomographic Locality, too. In fact, if $A$ and $B$ are classical state spaces of $M$ resp. $N$ outcomes, then $\Omega_{AB}^{\min} = \Omega_{AB}^{\max}$, and so there is a *unique* composite: the classical state space of $MN$ outcomes. It has recently been shown (Aubrun *et al.*, 2019 [3]) that this property characterizes classical state spaces: if $\Omega_{AB}^{\min} = \Omega_{AB}^{\max}$ then necessarily one of $A$ or $B$ must be classical, i.e. $\Omega_A$ or $\Omega_B$ must be a simplex.

What is more, composite state spaces automatically satisfy the **no-signalling principle**. In this sense, GPTs are generalizations of QT that avoid some of the problems of ad-hoc modifications discussed in Section 2.

**Lemma 18** (No-signalling principle). *Bell scenarios as in Figure 2 are modelled in GPTs with the prescription*

$$P(a,b|x,y) = e_x^{(a)} \otimes e_y^{(b)}(\omega_{AB});$$

*that is, $\omega_{AB}$ represents the initial global preparation procedure on the composite state space (see Definition 16); for every choice of input $x$ for Alice (resp. $y$ for Bob) there is a corresponding measurement $\{e_x^{(a)}\}_a$ with outcomes labelled by $a$ (resp. a measurement $\{e_y^{(b)}\}_b$ with outcomes labelled by $b$), and the local measurements are performed independently. These probability tables satisfy the no-signalling principle.*

*Proof.* This is very easy to demonstrate: note that the effects of any measurement sum up to the normalization functional, hence, by linearity,

$$\sum_b P(a,b|x,y) = \left( e_x^{(a)} \otimes \sum_b e_y^{(b)} \right)(\omega_{AB}) = e_x^{(a)} \otimes u_B(\omega_{AB}).$$

This is manifestly independent of $y$. An analogous argumentation can be applied with Alice and Bob interchanged, showing that $P(a,b|x,y)$ satisfies the no-signalling conditions. □

This calculation can also be used to define a **local reduced states** that generalize the partial trace of quantum mechanics:

$$\omega_A = \mathbf{1}_A \otimes u_B(\omega_{AB}), \quad \omega_B = u_A \otimes \mathbf{1}_B(\omega_{AB}).$$

There are some further intuitive and less trivial consequences of the definition of a composite in GPTs. For example, if $\omega_A$ and $\omega_B$ are both pure states then so is $\omega_A \otimes \omega_B$. This can be shown by considering the local *conditional states*, see Janotta and Hinrichsen 2014 [47]. Note however that this property fails in general if the principle of Tomographic Locality is not assumed, see e.g. Barnum *et al.*, 2016 [7].

Another simple consequence of the definition is that the *capacity is supermultiplicative* (recall Definition 10):

**Lemma 19.** *For any composite $(AB, \Omega_{AB})$ of two state spaces $(A, \Omega_A)$ and $(B, \Omega_B)$, it holds $N_{AB} \geq N_A \cdot N_B$.*

*Proof.* Let $\omega_1^A, \ldots, \omega_{N_A}^A$ be some maximal set of perfectly distinguishable states in $\omega_A$, and $e_A^{(1)}, \ldots, e_A^{(N_A)}$ be the corresponding measurement that distinguishes these states. Similarly, let $\omega_1^B, \ldots, \omega_{N_B}^B$ be some maximal set of perfectly distinguishable states in $\Omega_B$, and $e_B^{(1)}, \ldots, e_B^{(N_B)}$ be the corresponding measurement. Then

$$e_A^{(i)} \otimes e_B^{(j)}(\omega_k^A \otimes \omega_l^B) = \delta_{ik}\delta_{jl} = \delta_{(ij),(kl)},$$

hence the $(N_A N_B)$ product states $\omega_k^A \otimes \omega_l^B \in \Omega_{AB}$ are perfectly distinguishable. □

In the final example, we will see how the GPT framework reproduces some of the beyond-quantum phenomena that we have discussed in Section 2: it admits superstrong nonlocality.

**Example 20** (Composition of two gbits)**.** *Let $(A, \Omega_A)$ and $(B, \Omega_B)$ be the gbit state spaces defined in Example 11. Consider the maximal tensor product of these two state spaces, $(AB, \Omega_{AB}^{\max})$.*

*Since $A = B = \mathbb{R}^3$ and $AB = A \otimes B$, this shows that $\Omega_{AB}^{\max}$ is an eight-dimensional compact convex set. What is this set? By definition, the maximal tensor product contains* all *vectors that give non-negative probabilities on the product measurements (and normalization is automatic). Recall Lemma 18: these probabilities are nothing but the probability tables in a Bell experiment. Now, as we have seen in Example 11, there are only two "pure" measurements of the gbit, which we have denoted $(e^{(x)}, \bar{e}^{(x)})$ and $(e^{(y)}, \bar{e}^{(y)})$. Thus, it is sufficient to check non-negativity for these two possible local measurements, which yield binary outcomes.*

*But this leads us to conclude that the states in $\Omega_{AB}^{\max}$ are in linear one-to-one correspondence to the set of all non-signalling $(2,2,2)$-probability tables. In other words, we conclude that* **the maximal tensor product of two gbits is equivalent to the no-signalling polytope of Figure 3.** *And this argumentation can indeed be made rigorous by slightly more careful mathematical formalization.*

*In particular, PR-boxes are valid states on AB. Actually, if we defined an entangled state $\omega_{AB}$ as a state that cannot be written as a convex combination of product states, i.e. $\omega_{AB} \in \Omega_{AB} \setminus \Omega_{AB}^{\min}$, then PR-boxes are entangled states, similarly as the singlet is an entangled state in QT.*

We can say more about this composite state space $\Omega_{AB}^{\max}$. In Subsection 3, we have claimed that the no-signalling polytope has 24 extremal points, including 8 versions of the PR-box. While the latter claim is somewhat cumbersome to verify, we can now easily understand the role of the remaining 16 extremal points: these are the pure product states. As illustrated in Figure 9, a single gbit has four pure states $\omega_1, \ldots, \omega_4$. Therefore, $\Omega_{AB}^{\max}$ must contain the 16 pure product states $\{\omega_i^A \otimes \omega_j^B\}_{i,j=1,\ldots,4}$.

The construction above can be generalized in an obvious way to more than two parties, and also to local systems with more than two pure measurements and two outcomes (Barrett 2007 [11]). What would our world look like if it was described by this kind of theory (sometimes colloquially called "boxworld") instead of QT? For example, what kind of reversible transformations would be possible? While QT admits a large group of reversible transformations (the unitaries), it can be shown that boxworld admits **only trivial reversible transformations**: local operations and permutations of subsystems. In particular, no correlations can be reversibly created, and no non-trivial computation can ever be reversibly performed (Gross *et al.*, 2010 [39]). It also means that no reversible transformation can map a pure product state to a PR-box state. This is in contrast to QT, where we can certainly engineer unitary time evolutions that map a pure product state to, say, a singlet state. In some sense, entanglement in a boxworld universe would represent a scarce resource which cannot be regained reversibly once it is spent.

**Further reading.** We have restricted our considerations to compositions of *pairs* of state spaces, and to *tomographically local* composites. In this case, there is an obvious list of requirements for composition, and we have incorporated all these requirements in Definition 16: product states and product measurements should be possible, and (as enforced by the tensor product rule) independent local operations should commute. In general, however, we may be interested in multipartite systems similar to the circuit model of quantum computation. There, it becomes very cumbersome to work out the set of constraints that arise from the multipartite structure. In this case, **category theory** becomes the tool of choice, see (Coecke and Kissinger, 2017 [27]) for an introduction. Moreover, some of the abstract linear algebra and convex geometry formalism can be traded for a more picturesque *diagrammatic* formalism which allows to prove results in QT and beyond in particularly intuitive ways, see e.g. Chiribella *et al.*, 2010 [19]. As an example, the old problem of how to deal with the tensor product of *quaternionic* quantum systems (which also falls into the GPT framework) can be resolved by

constructing dagger-compact categories of such systems, in which case composition becomes well-defined and well-behaved (Barnum *et al.*, 2016 [7]).

# 4 Quantum theory from simple principles

After this formal tour de force, we are ready to understand how QT can be derived from some simple physical or information-theoretic principles. This section sketches one such possible derivation published by Masanes and Müller, 2011 [56]. It relies on axioms that have first been written down by Hardy (2001) [40]. However, there are many alternative routes. Please see the "Further reading" paragraph at the end of this section for an overview.

As before, we will restrict ourselves to finite-dimensional state spaces, we will work under the "no-restriction hypothesis", and we will assume the principle of Tomographic Locality (see Definition 16). In addition, we will postulate two further principles:

- **Subspace Axiom**. For every $N \in \mathbb{N}$, there is a dynamical state space $(A_N, \Omega_N, \mathcal{T}_N)$ of capacity $N$ (e.g. for $N = 2$ a bit, for $N = 3$ a trit etc.) Moreover, if $e^{(1)}, \dots, e^{(N)}$ is any measurement that perfectly distinguishes $N$ states in $\Omega_N$, then the subset of states $\omega$ with $e^{(N)}(\omega) = 0$ is equivalent to $\Omega_{N-1}$, and the subset of transformations $T \in \mathcal{T}_N$ that preserve this subset is equivalent to $\mathcal{T}_{N-1}$.

- **Continuous Reversibility**. The group of reversible transformations $\mathcal{T}_N$ is connected ("continuous"), and for every pair of pure state $\varphi, \omega \in \Omega_N$ there is some reversible transformation $T \in \mathcal{T}_N$ that maps one state to the other, i.e. $T\omega = \varphi$.

In all of the rest of this section we will assume that these two principles hold.

The principle of Continuous Reversibility expresses the physical intuition that time evolution should be continuous and reversible, and that "all pure states are equivalent" under such time evolution. The Subspace Axiom is particularly well-motivated from scientific practice: whenever we claim to have a quantum bit (or even a classical bit) in the laboratory, then this bit is not a free-floating stand-alone system, but it is embedded into a larger system (the rest of the world). Our description of the *state space of the bit*, and of its reversible transformations, should be independent of the rest of the world, and in particular independent of other zero-probability events.

For example, if we have a three-level atomic system in the laboratory, and we are sure (say, due to constraints arising from the experimental setup) that we will never find the particle in the third level, then we should be able to treat this system as a two-level system. In the notation above, the three-level system would be described by some state space $\Omega_3$ (the $3 \times 3$ density matrices in the quantum case). There would be some measurement $e^{(1)}, e^{(2)}, e^{(3)}$ that perfectly distinguishes the three levels (the measurement operators $|1\rangle\langle 1|, |2\rangle\langle 2|, |3\rangle\langle 3|$ in the quantum case), and the set of states

$$\tilde{\Omega}_2 := \{\omega \in \Omega_3 \mid e^{(3)}(\omega) = 0\}$$

would be equivalent to a two-level system (the density matrices $\rho = \begin{pmatrix} \tilde{\rho} & 0 \\ 0 & 0 \end{pmatrix}$ with $\tilde{\rho}$ a $2 \times 2$ density matrix in the quantum case).

It turns out that these principles are sufficient to uniquely determine the quantum state space. In this section, we will sketch a strategy to reconstruct QT from these postulates alone. This is quite remarkable, given that neither the GPT framework nor any of the postulates makes use of any mathematical elements that are typically considered to constitute the basic structure of QM: complex numbers, wave functions, operators, or multiplication (a priori, GPTs do not carry any algebraic structure). That is, we will show the following theorem:

**Theorem 21.** *In the framework described above (which presumes Tomographic Locality), under the Subspace Axiom and the postulate of Continuous Reversibility, the dynamical state space of capacity $N$ must be equivalent to $(A_N^{\mathrm{QT}}, \Omega_N^{\mathrm{QT}}, \mathcal{T}_N^{\mathrm{QT}})$, where*

- $A_N^{\mathrm{QT}} = \mathbf{H}_N(\mathbb{C})$ *is the real vector space of Hermitian $N \times N$-matrices,*

- $\Omega_N^{\mathrm{QT}} = \{\rho \mid \mathrm{tr}(\rho) = 1, \rho \geq 0\}$ *is the set of $N \times N$ density matrices,*

- $\mathcal{T}_N^{\mathrm{QT}} = \{\rho \mapsto U\rho U^\dagger \mid U^\dagger U = \mathbf{1}\}$ *is the group of unitary conjugations.*

Arguably, we can thus see the resulting reconstruction as some sort of explanation for "why" QM has these counterintuitive structural elements in the first place. It also reinforces the earlier insight that QM is in some sense a very "rigid" theory which is hard to modify without breaking some cherished physical principles. For more discussions on this, see e.g. Koberinski and Müller, 2018 [52], and the references therein.

We will proceed in a couple of steps. Our first step will be to understand why the quantum bit (or, rather, any capacity-two-system $\Omega_2$ that satisfies the postulates) must be equivalent to a Euclidean ball. This will also provide a quite illuminating explanation for the Bloch ball and its properties.

## 4.1 Why is the qubit described by a Bloch ball?

Let us begin with the most boring and trivial case: $\Omega_1$, the state space with capacity $N = 1$. In the quantum case, this corresponds to the $1 \times 1$ density matrices – containing only the trivial state $\rho = (1)$ on the trivial Hilbert space $\mathbb{C}^1$. But we do not know this yet, so let us *only* work with the postulates above.

**Lemma 22.** *The state space of capacity $N = 1$ is equivalent to the trivial state space $(\mathbb{R}, \{1\})$. In other words, $\Omega_1$ contains only a single state.*

We will not formally prove this lemma, but instead give some intuition for why it is true. Consider any state space $(\mathbb{R}^d, \Omega)$ (with $d \in \mathbb{N}$ arbitrary) that contains *more* than one state. If $\varphi_1, \varphi_2$ are two different states in $\Omega$, then the line segment $\{\lambda\varphi_1 + (1-\lambda)\varphi_2 \mid 0 \leq \lambda \leq 1\}$ is also contained in $\Omega$, hence $\dim \Omega \geq 1$, and so $d \geq 2$.

Now, convex geometry (Webster 1994) tells us that we can always find some pure state $\omega_1 \in \Omega$ that is an *exposed point*: namely, there is a hyperplane $H_1$ (of dimension $d-1$) that touches $\Omega$ in exactly that point:

$$H_1 \cap \Omega = \{\omega_1\}.$$

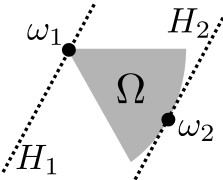

Figure 10: State spaces $\Omega$ that contain more than one state have capacity at least $N = 2$ (for argumentation see main text). The plane here is the affine space $\{x \in A \mid u_A(x) = 1\}$.

Since $\Omega$ contains more than one state, there must be other states of $\Omega$ that are not contained in $H_1$, but that are contained in hyperplanes parallel to $H_1$. In particular, there will be one such hyperplane (call it $H_2$) that touches $\Omega$ on the "opposite side" as depicted in Figure 10.

Hence all of $\Omega$ is contained in $H_1$ and $H_2$ and in between, and $H_2 \cap \Omega$ is not empty. Pick now any state $\omega_2$ from this intersection.

Now every hyperplane is the level set of an affine functional (which becomes linear if we add in the normalization degree of freedom). That is, we can find some linear functional $e^{(1)} \in (\mathbb{R}^d)^*$ such that $e^{(1)}(x) = 1$ for all $x \in H_1$ and $e^{(1)}(x) = 0$ for all $x \in H_2$. Since all $\omega \in \Omega$ lie in between the two hyperplanes, we have $0 \le e^{(1)}(\omega) \le 1$ for all $\omega \in \Omega$. Thus, $e^{(1)}$ is a valid effect (recall that we assume the no-restriction hypothesis in all of these lecture notes; otherwise, we would need an additional argument to show that $e^{(1)}$ is physically allowed). Define $e^{(2)} := u - e^{(1)}$, where $u$ is the normalization functional. Then $(e^{(1)}, e^{(2)})$ constitutes a measurement that perfectly distinguishes the two states $\omega_1$ and $\omega_2$. Therefore the capacity is $N \ge 2$.

This shows that state spaces with capacity $N = 1$ contain only a single state.

Next, let us use similar reasoning to say something slightly more interesting:

**Lemma 23.** *The state space $\Omega_2$ of capacity $N = 2$ is* strictly convex, *i.e. does not contain any lines in its boundary.*

Again, we will not really give a formal proof, but appeal to geometric intuition. Suppose that $\Omega_2$ was *not* strictly convex. Then, with a similar construction as above, we could find some hyperplane $H_1$ that touches $\Omega_2$ in more than one point, see Figure 11. Pick any $\omega_1 \in H_1 \cap \Omega$. Furthermore, let $H_2$ be the "opposite" hyperplane, and pick some $\omega_2 \in H_2 \cap \Omega$. As above, we can associate a measurement $(e^{(1)}, e^{(2)})$ to these two hyperplanes that perfectly distinguishes $\omega_1$ and $\omega_2$.

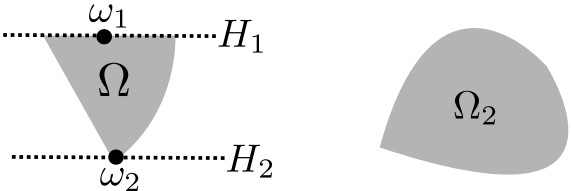

Figure 11: Assuming the Subspace Axiom, the bit state space $\Omega_2$ must be strictly convex, i.e. cannot contain lines in its boundary like the convex set depicted on the left. Instead, it could look like the convex set on the right.

Let us now invoke the Subspace Axiom. It tells us that the set

$$\{\omega \in \Omega_2 \mid e^{(2)}(\omega) = 0\} = \{\omega \in \Omega_2 \mid e^{(1)}(\omega) = 1\} = H_1 \cap \Omega$$

must be linearly equivalent to $\Omega_1$. But this set contains infinitely many states, whereas $\Omega_1$ contains only a single state. This is a contradiction.

We thus conclude that $\Omega_2$ must roughly look like the convex set in the right of Figure 11. Formally, this means that all of its boundary points must be pure states. Let us now additionally invoke the postulate of Continuous Reversibility and show the following:

**Lemma 24.** *The state space $\Omega_2$ is equivalent to a Euclidean unit ball of some dimension.*

In other words, we will now derive the fact that a quantum bit is described by the Bloch ball. However, we will not (yet) be able to say that this ball must be three-dimensional.

Let us start by defining what one may call the "maximally mixed state" of $\Omega_2$: pick any pure state $\omega \in \Omega_2$, and define $\mu := \int_{\mathcal{T}_2} T\omega \, dT$; that is, we integrate over the invariant (Haar) measure of the group of reversible transformations $\mathcal{T}_2$ (group averaging). It follows that $T\mu = \mu$ for all $T \in \mathcal{T}_2$, and it is easy to check that $\mu$ is in fact the *unique* state with this property.

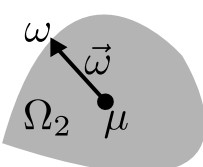
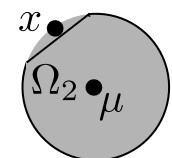

Figure 12: Left: The definition of Bloch vectors embeds the normalized states into a linear space (of one dimension less than the linear space on which the state cone lives). Right: If any point on the sphere does *not* correspond to a valid state, then this contradicts the strict convexity of $\Omega_2$.

For states $\omega \in \Omega_2$, we define the corresponding "Bloch vector" $\vec{\omega} := \omega - \mu$ (see Figure 12). Hence, $T\omega = \varphi$ if and only if $T\vec{\omega} = \vec{\varphi}$, and $\vec{\mu} = 0$. Then $\mathcal{T}_2$ acts on the linear space that contains the Bloch vectors. Now we can use a well-known trick from group representation theory (Simon 1996 [76]), and construct an *invariant inner product*. Namely, if "·" is an arbitrary inner product on the space of Bloch vectors, then we can define

$$\langle \vec{x}, \vec{y} \rangle = \alpha \int_{\mathcal{T}_2} (T\vec{x}) \cdot (T\vec{y}) \, dT,$$

where $\alpha > 0$ is some normalization constant to be fixed soon. It follows that $\langle T\vec{x}, T\vec{y} \rangle = \langle \vec{x}, \vec{y} \rangle$ for all $T \in \mathcal{T}_2$. This tells us that we can choose coordinates in the Bloch space such that the $T$ are orthogonal matrices. Moreover, if $\omega$ and $\varphi$ are arbitrary pure states, then, due to Continuous Reversibility, there is some transformation $T \in \mathcal{T}_2$ such that $T\omega = \varphi$. Thus

$$\|\vec{\varphi}\|^2 = \langle \vec{\varphi}, \vec{\varphi} \rangle = \langle T\vec{\omega}, T\vec{\omega} \rangle = \langle \vec{\omega}, \vec{\omega} \rangle = \|\vec{\omega}\|^2.$$

The Bloch vectors of all pure states have the same Euclidean length, and we can fix it to $\|\vec{\varphi}\| = 1$ by a suitable choice of $\alpha$. Hence, *all pure states lie on the unit sphere surrounding $\mu$*, see Figure 12 right. Could there be any states on the sphere which do not correspond to pure states? This is only possible if the topological boundary of $\Omega_2$ contains lines, contradicting the strict convexity of $\Omega_2$.

## 4.2 Why is the Bloch ball three-dimensional?

We have now reconstructed the Bloch ball representation of a qubit, but not its dimensionality. If the dimension of the bit state space is $d = 1$, then we recover an old friend from Figure 7: the classical bit. But, as we have seen in Subsection 3.2, composing classical bits will give us classical state spaces with a *discrete* (not connected) group of reversible transformations. The Bloch ball dimension must thus be $d \geq 2$.

We can say more about its dimension by considering *composites of several bits*.

The first step is to prove that the *capacity is multiplicative*:

$$N_{AB} = N_A N_B. \tag{6}$$

This follows from two lemmas that are proven by making use of all the postulates (see the paper by Masanes and Müller, 2011 [56] for details): first, that the maximally mixed state composes as $\mu_{AB} = \mu_A \otimes \mu_B$; and second, that the maximally mixed state on any system $A$ can be written $\mu_A = \frac{1}{N_A} \sum_{i=1}^{N_A} \omega_i^A$, where $\omega_1^A, \ldots, \omega_{N_A}^A$ is a maximal set of pure and perfectly distinguishable states of $A$. In light of Eq. (6), we can now view the dimension $K$ of the state

space as a function of the capacity $N$. As first argued by Hardy (2001), the fact that $K$ is also multiplicative on composition, i.e. $K_{AB} = K_A K_B$, enforces that we must have the relation

$$K = N^r, \qquad (7)$$

where $r \in \mathbb{N}$ is some integer. This fits quite nicely out observation after Definition 10: CPT has $K = N$ (i.e. $r = 1$) and QT has $K = N^2$ (i.e. $r = 2$). This suggests that the unknown exponent $r$ is somehow related to the "order of interference" of the corresponding theory as introduced in Subsection 2.2.

Since $\dim \Omega = K - 1$, the dimension $d$ of the Bloch ball must be one of

$$d = 2^r - 1 \in \{1, 3, 7, 15, 31, \ldots\}.$$

We have already excluded $d = 1$, and we would like to show that $d = 3$ is the unique possibility consistent with the postulates. To this end, let us consider the group of reversible transformations $\mathcal{T}_2$ of a single bit. What can we say about it? We know that it must be a compact connected group, and we also know that it must satisfy the principle of Continuous Reversibility: all pure states are connected by some reversible transformation. In other words, $\mathcal{T}_2$ *must act transitively on the sphere*.

What groups satisfy these requirements? In fact, for arbitrary ball dimensions $d \in \mathbb{N}$, there are many such groups. For example, for $d = 6$, it can be SO(6), SU(3) or U(3) (see Masanes et al., 2014, for a complete list). However, if $d$ is odd (as we have shown above) then the answer is pretty simple, but with a surprising twist:

- If $d \neq 7$ then we must have $\mathcal{T}_2 = \mathrm{SO}(d)$.

- If $d = 7$ then we either have $\mathcal{T}_2 = \mathrm{SO}(7)$ or $\mathcal{T}_2 = \mathrm{G}_2$, the exceptional Lie group.

For simplicity, let us in the following ignore the $\mathrm{G}_2$-case (how to treat this case, and all other details of the proof, can be found in the paper by Masanes and Müller, 2011 [56]). To understand why $d = 3$ follows from our postulates, we have to consider a *pair* of two bits. Due to Eq. (6), its state space $\Omega_{2,2}$ is equivalent to $\Omega_4$. Consider two perfectly distinguishable states $\omega_0$ and $\omega_1$ of a single bit (as points of the state space, they must lie on opposite sides of the $d$-dimensional Bloch ball), and two corresponding effects $e_0$ and $e_1$ with $e_i(\omega_j) = \delta_{ij}$. Then the four states $\omega_i^A \otimes \omega_j^B \in \Omega_{2,2}$ are perfectly distinguishable. Now consider the subset

$$\Omega_2' := \{\omega \in \Omega_{2,2} \mid e_0^A \otimes e_1^B(\omega) = e_1^A \otimes e_0^B(\omega) = 0\}.$$

This subset contains two of the product states, $\omega_0^A \otimes \omega_0^B \in \Omega_2'$ and $\omega_1^A \otimes \omega_1^B \in \Omega_2'$. Using the Subspace Axiom twice, it follows that $\Omega_2'$ is again equivalent to a bit – it also corresponds to a $d$-dimensional Bloch ball that somehow sits inside the joint vector space $AB = \mathbb{R}^{d+1} \otimes \mathbb{R}^{d+1}$. And it must contain at least one (actually, many) non-product pure states $\omega$ — these will be entangled states.

Returning to bit $A$, consider rotations $R \in \mathrm{SO}(d)$ that preserve the axis that connects $\omega_0$ and $\omega_1$; these will be rotations with $e_i \circ R = e_i$ for $i = 0, 1$. The subgroup of such $R$ is equivalent to $\mathrm{SO}(d-1)$. We can also perform such rotations on bit $B$. But since they preserve the two effects $e_i$, they also preserve the bit $\Omega_2'$:

$$R \otimes S(\Omega_2') = \Omega_2' \text{ for all } R, S \in \mathrm{SO}(d-1).$$

Now, $\Omega_2'$ spans a pretty small affine subspace (we can turn it into a linear subspace $L_2$ by substracting the maximally mixed state of $\Omega_2'$): we have $\dim L_2 = d$. On the other hand, $\mathrm{SO}(d-1) \otimes \mathrm{SO}(d-1)$, where each factor acts in its fundamental representation, is a pretty

large group. It acts on a large subspace $\mathbb{R}^{d-1} \otimes \mathbb{R}^{d-1}$ that sits inside $AB$. We have just seen that it must preserve the small $d$-dimensional subspace $L_2$. Is this possible at all?

The answer comes from group representation theory. It turns out that *the fundamental representation of* $\mathrm{SO}(d-1)$ *is complex-irreducible if* $d \geq 4$. Thus, it follows that $\mathrm{SO}(d-1) \otimes \mathrm{SO}(d-1)$, as a representation of two copies of this group, *also* acts irreducibly: but this implies that there *cannot* be any proper invariant subspaces like $L_2$. This rules out the possibility that $d \geq 4$.

Why is the case $d = 3$ different? This is due to the fact that $\mathrm{SO}(3-1)$ is an *Abelian* group. Hence all its irreducible representations must be one-dimensional, and so its acts *reducibly* on $\mathbb{C}^2$. This is the reason why the argumentation above does not apply in this case. We can hence summarize our finding with the following slogan:

> The Bloch ball of Quantum Theory is three-dimensional "because" $\mathrm{SO}(d-1)$ is non-trivial and Abelian only for $d = 3$.

There is a surprising twist to this insight. Garner et al., 2017 [35], consider a thought experiment in which the two arms of a Mach-Zehnder interferometer are described by a $d$-dimensional Bloch ball state space. They study the question for which $d$ this "fits into relativistic spacetime" (under some background assumptions), in the sense that relativity of simultaneity is satisfied. Under one set of assumptions, it turns out that only $d = 3$ is possible — and the reason is, once again, that $\mathrm{SO}(d-1)$ is only non-trivial and Abelian for $d = 3$. This is the same "mathematical reason" as above, but with a different physical interpretation: now $\mathrm{SO}(d-1)$ corresponds to "local phase transformations" that do not alter the global statistics, and commutativity of this group is enforced by relativity. This points to a fascinating interplay between information-theoretic and spacetime properties of QT; see also Müller and Masanes 2013 [59] and Dakić and Brukner 2013 [30] for further insights into this relation.

The $d = 7$ Bloch ball with its transitive group $\mathrm{G}_2$ appears as a curious special case. While the above argumentation shows incompatibility with our postulates also for this case, there was some hope for a while that one can construct a non-quantum composite state space of 7-balls that satisfies the principles of Tomographic Locality and Continuous Reversibility, but not the Subspace Axiom, see e.g. Dakić and Brukner, 2013 [30]. Unfortunately, this possibility has since been ruled out for the case of two bits in Masanes et al., 2014 [57]. However, it is not known whether such a construction might be possible in the case of $n \geq 3$ bits. While Krumm and Müller, 2019 [53], rule out such non-quantum state spaces for $\mathrm{SO}(d)$ with $d \neq 3$, it remains open whether there is a curious post-quantum $\mathrm{G}_2$-related theory on more than two bits.

### 4.3 How do we obtain the quantum state spaces for $N \geq 3$?

The next step is to show that the state space of $k$ bits, for any $k \geq 2$, is equivalent to the state space of $k$ quantum bits. We begin with the case $k = 2$. From the argumentation above, we know that the two-bit dynamical state space can be written as $(\mathbb{R}^4 \otimes \mathbb{R}^4, \Omega_4, \mathcal{T}_4)$, i.e. $\Omega_4$ is a 15-dimensional compact convex set.

We already know that the states and transformations of single bits are equivalent to those of the quantum bit. Let us use this fact to introduce an equivalent representation in the sense of Definition 14. Recall the linear map $L$ from Example 15, mapping the Bloch vector representation of a qubit to its density matrix representation. Let us now apply the invertible linear map $L \otimes L$ to map $\mathbb{R}^4 \otimes \mathbb{R}^4$ into $\mathbf{H}_2(\mathbb{C}) \otimes \mathbf{H}_2(\mathbb{C}) \simeq \mathbf{H}_4(\mathbb{C})$. In this representation, the elements of $\Omega_4$ become self-adjoint unit-trace matrices. We know that $\Omega_4$ contains all quantum product states and their convex combinations (the separable states), but we do *not* yet know that $\Omega_4$ is exactly the set of $4 \times 4$ density matrices.

To show that it is, we have to return to the previous subsection. The sub-bit $\Omega_2'$ contains the two antipodal product states $\omega_0^A \otimes \omega_0^B$ and $\omega_1^A \otimes \omega_1^B$, but there is a 2-sphere of pure states "in between". Mapping out the action of $SO(2) \otimes SO(2)$ on these states, and analyzing how the $SO(3)$-rotations of $\Omega_2'$ have to interact with those, one can show with some tedious calculations that these pure state must correspond to $|\psi\rangle\langle\psi|$ for $|\psi\rangle = \alpha|00\rangle + \beta|11\rangle$. Acting on those states via local rotations produces all pure quantum states of $\Omega_4$. Since we can similarly generate all quantum effects, and since these are full duals of each other, there cannot be any further states. This shows that $\Omega_4$ is equivalent to the two-qubit quantum state space. Moreover, the rotations that we have just described turn out to generate the full group of unitary conjugations.

If we now have $k \geq 3$ bits, then we can repeat the above argumentation for every pair among the $k$ bits. Since the unitary gates on *pairs* of qubits generate *all* unitaries, this implies that $\mathcal{T}_{2^k}$ must contain all unitary conjugations. These generate all quantum states and effects. Furthermore, any additional non-unitary transformation would map outside of the quantum state space.

This reconstructs QT for $N = 2^k$. For capacities that are not a power of two, we can simply invoke the Subspace Axiom to derive the quantum state space (and the group of unitary conjugations) also in this case. For example, $\Omega_3$ is embedded in the two-bit state space $\Omega_4$, and the Subspace Axiom tells us that it must have the form that we expect.

This concludes our proof of Theorem 21, and our reconstruction of finite-dimensional quantum theory.

**Further reading.** The search for alternative axiomatizations of QT dates back to Birkhoff and von Neumann (1936) [16]. It was followed by foundational work on Quantum Logic (Piron 1964) [68] as well as mathematical work on the characterization of the state spaces of operator algebras (Alfsen and Shultz, 2003 [1]) and several attempts to pursue a derivation of QT as above, for example in the operationally motivated work of Ludwig (1983) [55] and in the description of "relational quantum mechanics" by Rovelli (1996) [73]. The rise of quantum information theory has shifted the focus: it became clear that the main features of quantum theory are already present in finite-dimensional systems, and that the notion of composition plays an extraordinarily important role in its structure. This shift of perspective has led to a new wave of attempts to derive the quantum formalism from simple principles, pioneered by Hardy (2001) [40]. Despite the importance and ingenuity of Hardy's result, there remained some problems to be cured — in particular, one of the postulates from which he derived the quantum formalism was termed the "simplicity axiom", stating that the state space should be in some sense the smallest possible for any given capacity. In particular, this left open the possibility that there is in fact an infinite sequence of theories, characterized by the "order of interference" parameter $r$, see Eq. (7), and QT is just the $r = 2$ case. This was excluded ten years later, see Dakić and Brukner 2011 [29], Masanes and Müller 2011 [56], and Chiribella *et al.* 2011 [20] (see also d'Ariano, Chiribella, and Perinotti, 2017 [31]). These works gave complete reconstructions of the formalism of QT. A lot more progress and insights have been gained since then. For example, there is now a new reconstruction by Hardy (2011) [41] which does not make use of the Simplicity Axion, a diagrammatic reconstruction based on category theory (Selby *et al.*, 2018 [75]), a reconstruction "from questions", i.e. based on the complementarity structure of propositions (Höhn and Wever 2017 [44], and Höhn 2017 [42, 43]); there are several beautiful works by Wilce on deriving the more general Jordan-algebraic state spaces from the existence of "conjugate systems" resembling QT's maximally entangled states (e.g. Wilce 2017 [87]); and there is now a derivation of QT from *single-system* postulates only, namely spectrality and strong symmetry (Barnum and Hilgert, 2019 [8]), an immensely deep result that significantly improves on earlier work by Barnum *et al*, 2014 [9]. This list is far from complete, and it certainly excludes important work that does *not* fall into the GPT

framework but relies, for example, more on the device-independent formalism mentioned at the end of Subsection 2.1.

# 5 Conclusions

The framework of Generalized Probabilistic Theories (GPTs) yields a fascinating "outside perspective" on QT. It tells us that QT is just one possible theory among many others that could potentially describe the statistical aspects of nature. These theories share many features with QT, like entanglement or the no-cloning theorem (see Barnum et al., 2007 [5]), but they also differ in some observable aspects, e.g. in the set of Bell correlations that they allow, in the group structure of their reversible transformations, or in the interference patterns that they generate on multi-slit arrangements.

However, we have seen that QT is still special: it is the unique GPT that satisfies a small set of simple information-theoretic principles. These principles are formulated in purely operational terms, without reference to any of the mathematical machinery of QT like state vectors, complex numbers, operators, or any sort of algebraic structure of observables. Thus, reconstructing QT from such principles can tell us, in some sense, "why" QT has its counterintuitive mathematical structure.

These results give us arguably important insights into the logical structure of our physical world. But then, *what exactly* do they tell us? Can we learn anything about *how to interpret* QT, and about the nature of the quantum world? The hope for a positive answer to this question has been famously raised by Fuchs (2003) [33]. Fuchs' hope was that a reconstruction of QT would ground it in large parts on information-theoretic principles, *but not completely*. He wrote: *"The distillate that remains — the piece of quantum theory with no information theoretic significance — will be our first unadorned glimpse of 'quantum reality'. Far from being the end of the journey, placing this conception of nature in open view will be the start of a great adventure."*

However, the recent reconstructions, including the one summarized in these lecture notes, seem to have given us derivations of QT from *purely* information-theoretic principles, full stop. What do we make of this? At the conference "Quantum Theory: from Problems to Advances" in Växjö, 2014, Časlav Brukner argued as follows: *"The very idea of quantum states as representatives of information — information that is sufficient for computing probabilities of outcomes following specific preparations — has the power to explain why the theory has the very mathematical structure that it does. This in itself is the message of the reconstructions."* It is possible to acknowledge this beautiful insight while remaining completely agnostic about the problem of interpretation. Or one may contemplate a bolder possibility: perhaps our world is at its very structural bottom fundamentally probabilistic and information-theoretic in nature (Müller, 2020 [58])? Whatever this may mean, or whichever position one may want to take, information-theoretic reconstructions of QT can be a fascinating and enlightening piece of puzzle in the great adventure to make sense of our quantum world.

# Acknowledgments

I am grateful to the organizers of the Les Houches summer school on Quantum Information Machines 2019 — to Michel Devoret, Benjamin Huard, and Ioan Pop — for making this inspiring event possible. I would also like to thank the audience for stimulating and fun discussions during and after the lectures. This research was supported in part by Perimeter Institute for Theoretical Physics. Research at Perimeter Institute is supported by the Government of Canada through the Department of Innovation, Science and Economic Development Canada and by

the Province of Ontario through the Ministry of Research, Innovation and Science.

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
