# Peer review of "Probabilistic Theories and Reconstructions of Quantum Theory (Les Houches 2019 lecture notes)"

_SciPost Physics Lecture Notes, doi:SciPost Phys. Lect. Notes 28 (2021)_

## Round 2 · Referee Report · Anonymous (Referee 1) · 2021-3-11

Report

These lecture notes are about the framework of Generalized Probabilistic Theories (GPTs). This framework explores the set of possible physical theories (possibly going beyond quantum theory) using an operational approach. Notably, this framework allows to see what are (some of) the principles at the core of quantum theory. These possibilities make it an active field in quantum foundations.

The lecture notes start by motivating using such an approach that would go beyond quantum theory. This is followed by the introduction of the framework of GPTs, and finally by the reconstruction of quantum theory itself from L. Hardy's axioms.

While the topic is abstract and quite technical, the manuscript is very clear. The approach is rigorous and well motivated, and the mathematical proofs are given with the right amount of details for the physicist. This manuscript will also be an excellent entry point for the reader in the world of GPTs. Furthermore, the literature review exposed in the manuscript will allow the interested reader to deepen and go beyond the approach exposed in the manuscript. I therefore recommend its publication.

Requested changes

I have noted the following typos/minor mistakes/small questions during my reading: 1 - p. 4: Einstein has shown us that two simple physical principles single out […] 2 - p. 12: Chribella and Yuan 2016 -> Chiribella and Yuan 2016. 3 - p. 15: at contains enough dimensions -> it contains enough dimensions. 4 - p. 15: I think that the given the definition 5, the intersection of A_+ and -A_+ should be {0} and not the empty set. 5 - p. 18: Since the normalized states span […] 6 - p. 30: When discussing Lemma 22, it is mentioned that the effects e^{(1)} and e^{(2)} are physically allowed. It is not clear in the text whether it is a consequence of the no restriction hypothesis or not. 7 - p. 31: The justification of Eq. (6) does not appear obvious to me. In particular, without giving the details, what are the axioms/hypotheses that were used in order to obtain (6)? 8 - p. 33: irreps -> irreducible representations 9 - It could be worth mentioning some examples of what is achievable with the GPT framework beyond quantum theory. I am for example thinking about [C. Lee and J. Selby, Proc. R. Soc. A 2018 474 20170732 (2019)], in which the authors show that given some assumptions on the decoherence processes from a GPT to quantum theory, there are some limitations on "super quantum" theories.

---

## Round 3 · Author Response

I would like to thank the referee for their careful check of the manuscript and for their helpful comments. I have corrected all the typos, and I have implemented the following clarifications (the numbers are those given by the referee’s comments):

---

## Round 3 · List of Changes

6: After the half sentence on page 30, saying that e^{(1)} is a valid effect, I have aded the following comment in brackets: “(recall that we assume the no-restriction hypothesis in all of these lecture notes; otherwise, we would need an additional argument to show that e^{(1)} is physically allowed).”

7: I have expanded the explanation where Eq. (6) comes from; see now the top of page 32. Namely, it follows from two other lemmas: multiplicativity of the maximally mixed state, and representation of the maximally mixed state as a mixture of perfectly distinguishable pure states. All postulates are used to prove those. Since the lecture notes only intend to give a summary or sketch of the representation, I refer to our paper for the details.

9: I have added a reference to the paper by Lee and Selby on page 12. Note that, in the corresponding paragraph, I am also mentioning some other things that can be done with the GPT framework; in particular, to formulate consistent theories of higher-order interference.

---

## Editorial Decision

published